# *Van-der-Waals-forces*-modulated graphene-P-phenyl-graphene carbon allotropes

Huanxin Li [1,2,3] ✉, Haotian Chen [1], Boyi Pang[2,3], Jincan Zhang[4], Bingcheng Luo [4,5] ✉, S. Ravi P. Silva [6], Yi-Chi Wang [7], Siyu Zhao[3,8], Paul R. Shearing [3,8], James B. Robinson [2,3] ✉ & Kostya S. Novoselov [9,10] ✉

Graphene has received much attention due to its monoatomic, unique two-dimensional structure, which results in remarkable mechanical, physical, and electrical properties. However, synthesizing high-quality graphene-based composites with high conductivity and ionic mobility remains challenging. Here, we report an allotrope to the nanocarbon family, Graphene-P-phenyl-Graphene, synthesized by inserting π-π-conjugated p-phenyls between graphene layers and connecting them via C−C σ bonds. Graphene-P-phenyl-Graphene is thermally and dynamically stable, as verified by density functional theory and molecular dynamics, and can be produced at kilogram scale. The p-phenyl bridges swell the layer spacing from ~0.34 to ~0.56 nm, reducing van der Waals forces and enhancing electron delocalization. Electrons in these separated graphene layers benefit from low mass and efficient 3D screening of charge scattering, resulting in high Hall mobility (10,000–13,000 cm² V⁻¹ s⁻¹) in freestanding films. The expanded spacing also enables decoupling of layer electrons and rapid ion storage and transport−even for large ions. For example, potassium-ion batteries using Graphene-P-phenyl-Graphene exhibit high reversible capacity, long-term stability, and high charge-discharge rates. Graphene-P-phenyl-Graphene holds promise for large-scale, portable, high-performance electronics with energy storage capabilities.

Over the past thirty years, carbon allotropes such as fullerenes[1,2], carbon nanotubes[3], graphene[4] and their derivatives and composites have revolutionized our understanding of nanomaterials. The discovery of carbon allotropes has been recognized with three Nobel Prizes (the 1996 Nobel Prize in Chemistry for C60, the 2000 Nobel Prize in Chemistry for conductive polymers, and the 2010 Nobel Prize in Physics for graphene), and their impact continues to be far-reaching. A wide variety of carbon allotropes have been discovered or synthesized though (Fig. 1), to explore new carbon allotropes remains a highly challenging task[5].

Graphene, a single-atom-layer nanomaterial composed of six-membered carbon rings, exhibits high carrier mobility[6]. Single-layer graphene possesses extraordinary physical and chemical properties, such as the highest known in-plane conductivity in a material, but the production of high-quality, large-area single-layer graphene at scale still face significant technical challenges. Furthermore, the unique

[1]Department of Chemistry, Physical & Theoretical Chemistry Laboratory, University of Oxford, Oxford, United Kingdom. [2]Advanced Propulsion Lab, University College London, Marshgate, London, UK. [3]The Faraday Institution, Quad One, Becquerel Avenue, Didcot, UK. [4]Department of Engineering, University of Cambridge, Cambridge, UK. [5]College of Science, China Agricultural University, Beijing, China. [6]Advanced Technology Institute, University of Surrey, Guildford, Surrey, UK. [7]School of Materials Science and Engineering, Tsinghua University, Beijing, China. [8]Department of Engineering Science, University of Oxford, Oxford, OX, UK. [9]Department of Materials Science and Engineering, National University of Singapore, Singapore, Singapore. [10]Centre for Advanced Two-Dimensional (2D) Materials, National University of Singapore, Singapore, Singapore. ✉e-mail: huanxin.li@ucl.ac.uk; luobc21@cau.edu.cn; j.b.robinson@ucl.ac.uk; kostya@nus.edu.sg

physical and chemical properties of graphene, governed by the delocalization of π electrons, are significantly affected in multi-layer graphene or graphite-like materials due to the van der Waals interaction with adjacent layers. Few-layer graphene has attracted widespread attention due to the possibility to form topological electronic states, which generate optical features, and transport behaviour[7–10]. Control over the delocalisation of the π electrons between the layers is the key in governing such exciting phenomena, which is difficult to achieve in the traditional multilayer systems. By bridging the layers of graphene with appropriate π electron conjugated groups, it is possible to achieve overall control over the localization of π electrons. In addition, the modulation of the topological structure of few-layer graphene often brings unexpected surprises, such as superconductivity. Cao et al. initially discovered that the 1.1° magic angle in bilayer graphene superlattices could induce superconductivity[11], and recently, Zhou et al. reported superconductivity, "semi-metallic" and "quarter-metallic" characteristics in rhombohedral three-layer graphene[12,13]. Stevan et al. found evidence of unconventional superconductivity in twisted three-layer graphene[14]. These research results indicate that few-layer graphene has good electronic structure tunability and superconducting properties. The transport properties of its carriers (electrons, holes) can be modulated by regulating the delocalization state of the electrons between the layers, and under certain conditions, superconducting performance may be achieved.

Graphene-derived nanomaterials have been widely used in electrochemical energy storage materials due to their unique layered structure, stable physicochemical properties, and good electron transfer ability[15–17]. However, for ion migration in electrochemical processes, the van der Waals and electrostatic forces between graphene layers are important factors that restrict ion migration. Few-layer graphene is tightly bound together by van der Waals forces, with an interlayer spacing of only about 0.34 nm. Firstly, when ions are embedded between graphene layers, the interlayer spacing needs to be expanded to, which requires overcoming the van der Waals force, and this part is the insertion energy barrier of ions. Secondly, during the process of ion migration between graphene layers, electrostatic forces brought by the interlayer electronic cloud will affect ion migration. Since the van der Waals force is inversely proportional to the sixth power of the distance between the graphene layers ($Fv \propto R^{-6}$), while the coulombic force is inversely proportional to the square of the distance ($F_E \propto R^{-2}$), when the interlayer distance of graphene is small, these two charge derived forces make it difficult for ions to achieve rapid migration[18]. By rational design of the structure of few-layer graphene and expanding the interlayer spacing of carbon nanomaterials within a controlled range, the resistance of ion insertion and migration can be effectively reduced, and the energy barrier would be lowered. However, adjusting the interlayer spacing of graphene at the sub-nano metre level, especially achieving precise and controllable adjustment, is a very challenging task. Previous literature reports have shown that doping nanocarbon materials with heteroatoms can induce in-plane distortions and deformations in graphene, consequently altering the interlayer spacing. For example, carbon nanomaterials doped with elements such as nitrogen, sulphur, and phosphorus have a degree of expansion in interlayer spacing, and their capacity and rate performances are improved when used as negative electrode materials for ion batteries, with their cycling stability also improved[19–21]. In addition, the interlayer distance of graphene can be greatly increased by oxidizing the graphene material, and this method has been widely utilized in the chemical exfoliation of graphite carbon materials to prepare graphene[22]. Furthermore, the use of ion intercalation to embed charged ions/particles into the interlayer of graphene can also significantly widen its interlayer spacing. The method has been successfully used to prepare expanded graphite[23]. Although the above methods and approaches can change the interlayer spacing of graphene, they inevitably destroy the planar structure of graphene and

have a high degree of randomness in the process, making it difficult to achieve controllable adjustment of the interlayer and to carry out quantitative studies of its physicochemical properties that lack precise theoretical analysis models. To achieve controllable adjustment of the interlayer spacing of graphene nanomaterials, molecular and atomic-level structural design must be carried out.

Herein, we introduce a carbon allotrope, Graphene-P-phenyl-Graphene (GPG), formed by bridging π electron conjugate group, P-phenyl, into the interlayers of graphene. Our focus is to explore the electron/hole mobility and ion migration behaviours of GPG, as well as its potential applications in rapid rechargeable potassium ion batteries. To identify the stable and rational models for GPG, we utilized DFT calculations and MD simulations, which identified two types of GPG models, denoted as H type and Z type. To verify the structure models for GPG, we constructed GPG using high-quality single-layer CVD graphene. Importantly, we were able to produce GPG at kilogram-scale using a graphene oxide or liquid exfoliated graphene as a precursor, which resulted in a more defective structure but still maintained high-conductivity, enabling high-rate electron mobility and ion migration. Our structural analysis techniques, including Aberration-Corrected Transmission Electron Microscopy (ACTEM), Solid-State Nuclear Magnetic Resonance (SSNMR), Atomic Force Microscopy (AFM), and Raman spectroscopy, confirmed the structure of GPG. Furthermore, our DFT calculations indicated a lower energy barrier for potassium ion migration in GPG, suggesting its potential as a promising electrode material in ion batteries. We tested GPG as negative electrode material for potassium ion batteries and found it to exhibit extremely high-rate tolerance up to 210 C ($1 C = 300 \, mA \, g^{-1}$) and long-term stability for 20,000 cycles. We anticipate that GPG carbon allotropes will display more interesting properties and hold great potential for applications in batteries, sensors, catalysts, and other fields.

## Results and discussion
### Theoretical models
The quest for carbon allotropes other than graphene, C60, diamond, CNTs, graphenylene and biphenylene networks, has stimulated substantial research efforts because of the materials' predicted mechanical, electronic, and transport properties. However, their synthesis remains challenging given the lack of reliable protocols for scalable producing, especially at the kilogram-scale. (detailed in Supplementary Note 1). The existing allotropes formed from sp, sp² and sp³-hybridized carbon, including C18[4], biphenylene network[8], and long-range ordered porous carbon (LOPC)[5] as shown in Fig. 1a. Inspired by the flexibility of carbon allotrope forms, we report two graphene-p-phenyl-graphene (GPG), designated as Z type (Fig. 1-i) and H type (Fig. 1-ii), after searching the vast number of models (Supporting Information Fig. S3 and Table S2). The Z type GPG shows an angle of around 54.3 degree between graphene and p-phenyl planes, while the H type displays a vertical angle of 90.0 degree according to the optimized model (Fig. 1b), where the optimized top view and side view structures and Charge Density Difference (CHGDIFF) of Z type and H type GPG are illustrated. We calculated the average energy for each carbon in both Z and H types, which suggests the H type allotrope is the most stable system. The formation energies for various GPG types were illustrated (Fig. S4), indicating that Z type GPG is easy to form thematically. Additional DFT calculations were carried out using initial configurations with varying rotation and intersection angles (40°, 50°, 60°, and 70°) between the p-phenyl groups and the graphene plane (Fig. S17), confirming the thermodynamic preference for the formation of Z-type and H-type GPG structures. The band structures and Density of States (DOS) of both Z and H type GPG were demonstrated in Fig. 1c. It is interesting to notice that both the Z and H type GPG display a non-zero density of state at the Fermi level (opposite to graphene), offering enhanced conductivity. It is also worth mentioning that the DOS of H type GPG exhibit very sharp maxima, which might give rise to various

many-body effects. This is purely theoretical, with experimental verification yet to fellow. But we do record some high hall mobility (10000-13000 $cm^2V^{-1}s^{-1}$) at $25 \pm 1\,°C$ for the GPG films that could be contributed by such a feature (Fig. S2b). In addition to the idealized H- and Z-type models, we have explored additional configurations incorporating point defects and multilayer stacking, which likely coexist in real materials. These expanded models further confirm the structural stability and representative nature of the H- and Z-type motifs while highlighting the complexity and tunability of GPG materials (Figs. S15, S16). To elucidate the vibrational characteristics of the two GPG allotropes, we calculated the phonon density of states (DOS) and the phonon dispersion relations for both H-type and Z-type configurations, as presented in Fig. 1d. The DOS plots for the H-type and Z-type GPG structures reveal rich vibrational features spanning the entire frequency spectrum. In the H-type structure, a prominent broadening of phonon states is observed in the frequency window between approximately 1400 and $1750\,cm^{-1}$ (region **X**), where localized vibrational modes associated with the phenyl linkers are likely to emerge. This spectral congestion suggests enhanced phonon–phonon scattering and mode hybridization due to reduced symmetry and strong interfacial coupling between the phenyl units and the graphene layers. In contrast, the Z-type GPG also exhibits a distribution of vibrational states in the same spectral range (region **Y**), albeit with subtle differences in the peak positions and intensity distribution. These variations reflect the altered stacking geometry and phenyl rotation in the Z-type structure, which modulate the vibrational couplings and interlayer interactions. Phonon dispersion relations further support these observations. The H-type configuration displays noticeable band folding and mode clustering within the $1400$–$1750\,cm^{-1}$ range (panel **x**), indicating strong zone-folding effects and symmetry-lowering phenomena. Conversely, the Z-type dispersion (panel **y**) features a smoother band structure in the same frequency range, but with distinct mode splitting, highlighting the influence of torsional asymmetry and interlayer bonding heterogeneity. The vibrational landscape of GPG is highly sensitive to geometric configuration. The phonon signatures of the H- and Z-type GPG allotropes not only distinguish their structural motifs but also provide insight into their potential thermal transport and vibrational coupling behaviours.

Specifically, computer virtual screening was adopted to search for suitable bridging molecules, which result in several stable structures. The stable structures for p-phenyl bridged graphene were predicted by DFT calculations. (detailed in Supplementary Note 3). The most rational structures for p-phenyl bridging were donated as A0, A1, A2, B0, B1, B2, C0, C1 and C2, where 0, 1, 2 represent the number of covalent bonds that p-phenyl formed with graphene (see CIF files in **Source Data**). The average energy for each carbon suggests the C2 type allotrope is the most stable system. The formation energies and average energy per carbon were illustrated (Fig. S3), indicating that B2 type GPG is easy to form thematically. (detailed in Supplementary Note 4). For graphite, the van der Waals forces between the layers could be quantified by the *Equation* 1-3 **in SI**. The *Equation* 4−9 illustrate the relationship between the energy barrier for ions and layer spacing for layered materials. The findings demonstrate that larger layer spacing corresponds to lower energy barriers (*Equation* 10). This groundbreaking discovery highlights the immense potential of GPG—an innovative, flexible, controllable, and precisely engineered carbon nanomaterial—in various applications within nanotechnology and energy storage devices[24].

## Verification

According to the modelling results, here, we specifically propose an approach to realize the preparation of p-phenyl bridging graphene (GPG) by a diazotization reaction followed by further reduction. The detailed reaction process is illustrated in Supporting Information Fig. S1 (Supplementary Note 2). We initially selected the CVD graphene as substrate to fabricate a dual-layer GPG, as shown in Fig. 2a. The Cu substrate of single layer graphene was removed by a standard etch approach and transferred on the p-Phenylenediamine (PPD) solution. Then another layer of graphene on Cu was immersed into the solution and placed beneath the top one. After initiating the reaction between PPD and graphene with diluted HCl solution (0.05 M), the GPG precursors were obtained, and the final product was further washed with DI water and annealed under inert atmosphere to remove the residents. Afterwards, the GPG could be transferred onto different kinds of substrate (Si, $SiO_2$ etc.) depending on the requirements. The three-dimensional CHGDIFF of GPG obtained from simulation was shown in Fig. 2b, suggesting the introducing of p-phenyl aroused the charge redistribution both inside the layers and on the graphene planes. The more detailed charge redistribution for the interlayer and graphene plane were illustrated in Fig. 2c and Fig. S6 (detailed in Supplementary Note 5). The ACTEM image for the GPG produced via CVD graphene was shown in Fig. 2d, indicating that a certain degree of defects was generated on the graphene plane. The AFM image in Fig. 2e confirmed the existence in-plane defects, as well as some nanoholes (detailed in Supplementary Note 7). We measured the distance of the mono/bilayer edge from the top to the bottom graphene to evaluate the layer spacing of GPG ($0.59 \pm 0.10\,nm$), which is well-matched with the optimized model by DFT (Fig. 2f). Following the TEM image of GPG synthesized from CVD graphene (Fig. 2g), the enlarged ACTEM image (Fig. 2h) provides a clearer view of the atomic structure of H-type GPG (detailed in Supplementary Note 8 and 9). The region highlighted by the red square is further magnified in the inset, revealing a well-defined hexagonal atomic lattice. This lattice matches precisely with the top-view structural model of H-type GPG (Fig. 2i), where the atomic positions exhibit good alignment. The corresponding side-view model (Fig. 2j) confirms that the regular hexagonal lattice arises from the top graphene layer (highlighted in pink), while the atoms observed at the centre and periphery of the structure (in lilac) originate from the upper shoulder atoms of the p-phenyl groups.

## Scalable synthesis

According to the modelling and verification results, we specifically use an approach to realize the massive production of GPG at kilogram scale via a diazotization reaction followed by further reduction (Fig. S1). The kilogram-scale GPG precursor were achieved by diazotization reaction starting with chemically oxidized graphene oxide (GO, Fig. 3a) or exfoliated graphene flakes (Fig. 3b), both of which were produced by wet chemical process and is easily scaled. (detailed in Supplementary Note 6). The massive GPG precursors could be further reduced to be flexible carbon film (Fig. 3c) or powder under inert atmosphere. The macroscopic 3D morphology of the fabricated GPG film was obtained using X-ray micro-Computed Tomography (CT) (Fig. 3d), which depicts a distribution of pores in the carbon film in 3D space. After extracting the pore phase and carbon film phase according to the grey scale level using thresholding segmentation, the skeleton of the active materials can be visualised, from which a complex 3D structure and broad distribution of the pore size are observed. The effect of the P-phenyl introduction is obvious, as indicated by the HRTEM images. The multiple layered graphene displays an ordered horizontal side view with a layer spacing of 0.34 nm (Fig. 3e). While the GPG shows an interlayer with larger spacing and more disordered structures (Fig. 3f). The enlarged image (Fig. 3g) demonstrates the hexahydric ring of carbon which is the p-phenyl structure in the interlayers. For the top view of the GPG, the Z-type allotropes dominate the kilogram-scalable-synthesized GPG (Fig. 3h). The HRTEM image (Fig. 3i) on the top view of GPG shows a plane defect derived from reconstruction and deformation of carbon atoms due to the bonding of p-phenyl. The QSTEM simulation well-match the HRTEM image, which was obtained from quantitative simulation of the Z-type GPG model. Additionally, the

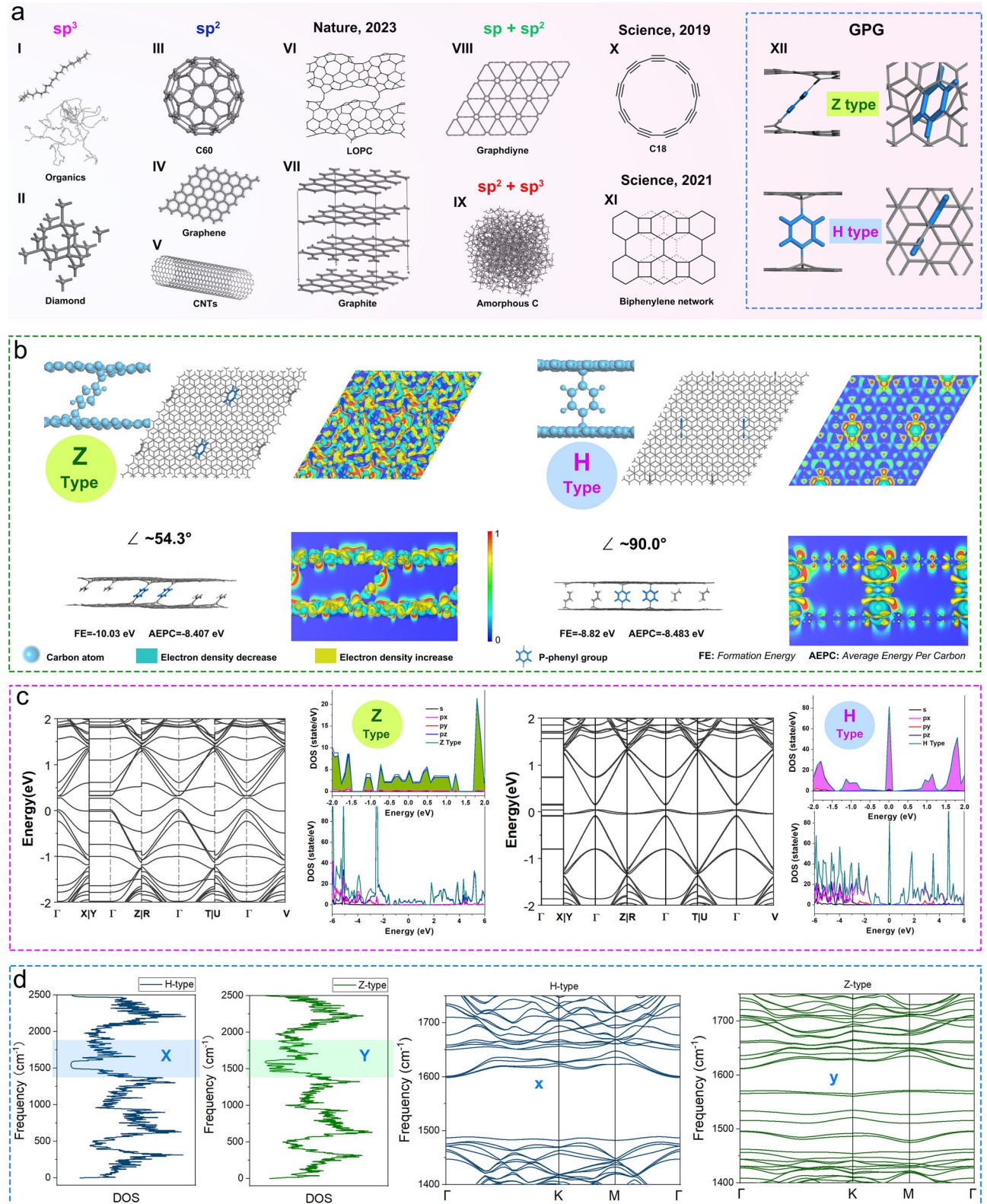

**Fig. 1 | The atomic structures of various carbon allotropes and band structures and corresponding electron and phonon DOS for Z-type and H-type GPG.** **a** Different types of carbon allotropes with sp, sp[2] and sp[3]-hybridized (a-I: organic carbon skeleton; a-II: diamond; a-III: C60; a-IV: Graphene; a-V: CNTs; a-VI: LOPC; a-VII: Graphite; a-VIII: Graphdiyne; a-IX: Amorphous Carbon; a-X: C18; a-XI: Biphe-nylene network; a-XII: Z-type GPG (top) and H-type GPG (bottom)); **b** The structures and Charge Density Difference (CHGDIFF) of Z- and H-type GPG allotropes (Blue ball: carbon atom; Grey stick structure: C-C bond; blue stick structure: highlighted p-phenyl group; Green area in CHGDIFF image: electron density decrease; yellow area in CHGDIFF image: electron density increase; the electron density increases relatively from deep blue (minimal, 0) to red (maximum, 1) on the colour bar); **c** The band structures and corresponding electron DOS of Z-type (left) and H-type (right) GPG allotropes; **d** The phonon density of states and dispersion profiles of Z- and H-type GPG allotropes (First: phonon DOS of H-type GPG; Second: phonon DOS of Z-type GPG; Third: the phonon dispersion of H-type GPG in region X; Fourth: the phonon dispersion of Z-type GPG in region Y).

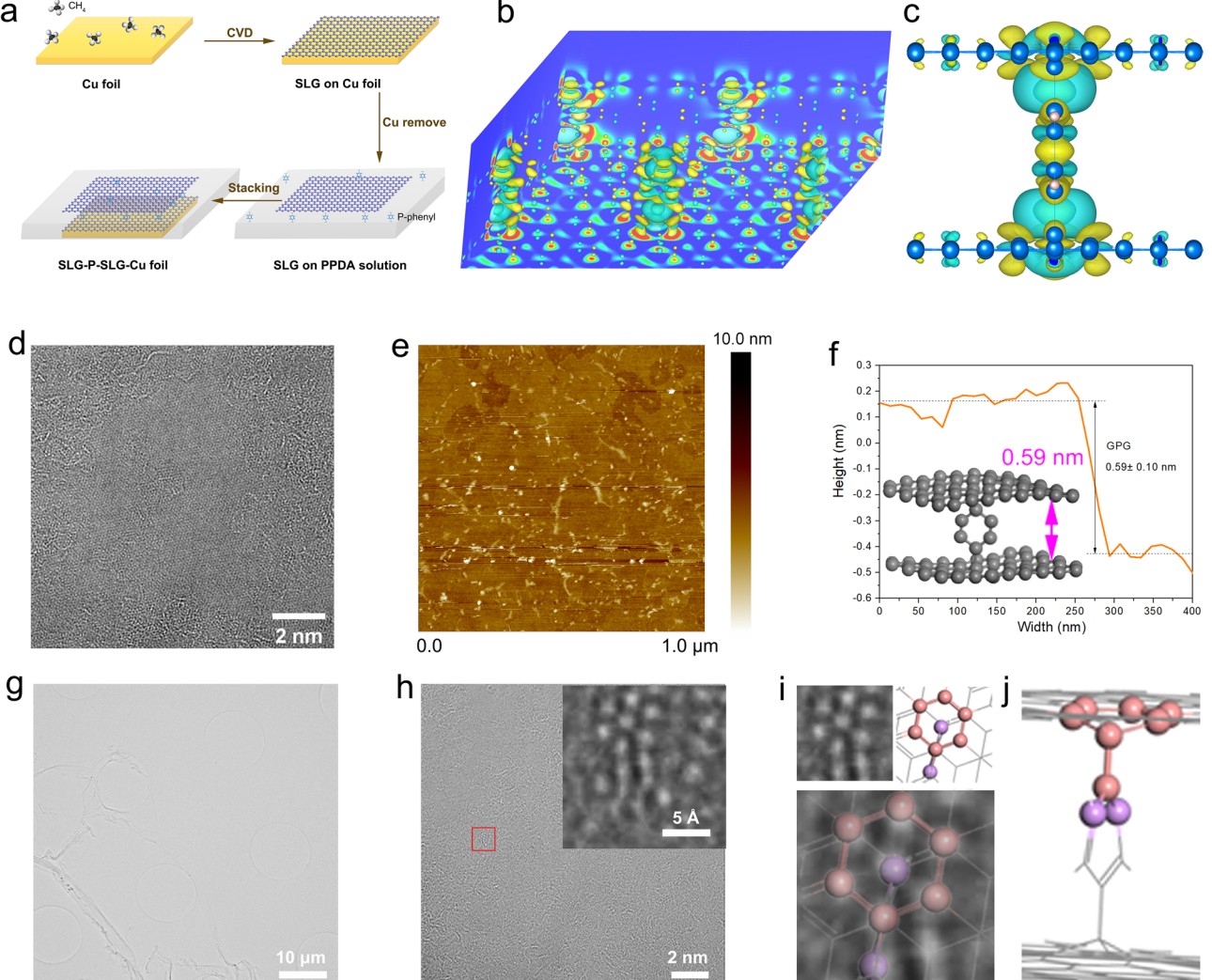

**Fig. 2 | Synthesis and characterizations of H-type GPG. a** The schematic for layer-by-layer synthesis of GPG from high quality CVD graphene; **b** The 3D and (**c**) detailed side view CHGDIFF of GPG between the layers; **d** ACTEM image of GPG top view; **e** AFM image of GPG top view; **f** The hight profile for single/double-layer GPG edge (grey ball are carbon atoms); **g** TEM image of GPG; **h** Enlarged TEM and ACTEM (inserted) images of GPG; **i** ACTEM image and modelling structure of the top view for H-type GPG: the original ACTEM image (top left); the theoretical model of of H-type GPG (top right); the alignment of ACTEM image and theoretical model of H-type GPG (bottom); **j** Side view of the modelling structure of H-type GPG (The grey sticks represent C–C bonds. All balls denote carbon atoms, with the pink and purple ones corresponding to the same set of carbon atoms in the top view (**i**) and side view (**j**), respectively. The ball-shaped atoms are those visible in the top view and are highlighted accordingly).

CHGDIFF of the Z-type GPG indicates that the introduction of p-phenyl significantly alters the charge distribution on graphene plane, which is beneficial to its high mobility. The typical TEM image of GPG shows a relatively rough surface (Fig. 3j) compared to the multiple-layered graphene. Raman spectrum of Z-type GPG demonstrates the D, G and 2D peaks (Fig. 3k). The D peak (1350-1360 cm$^{-1}$) in the Raman spectrum of graphene corresponds to a specific Raman scattering process associated with defects or disorder in the carbon lattice. The G peak (normally around 1580 cm$^{-1}$) corresponds to the in-plane vibrational mode of the sp$^2$ carbon atoms in the graphene lattice. The 2D peak (around 2700 cm$^{-1}$) arises due to the double-resonance process involving two phonons, which involves two non-identical phonons with opposite momentum. The typical G peak (Fig. 3l) split into two peaks (G around 1590 cm$^{-1}$, and G$^+$: 1605 cm$^{-1}$) which is ascribed to the molecular doping and trace N-type doping (detailed in Supplementary Note 10 and 11).

## Properties

As expected, the GPG pyrolysis at 1600 °C (GPG-p-1600) possesses a high hall mobility (10,000–13,000 cm$^2$V$^{-1}$ s$^{-1}$ at 25 ± 1 °C, which is very close to the theoretical value of single-layer graphene), as well as high conductivity (up to 10$^5$ S m$^{-1}$). We explore the potential of GPG for rapid chargeable batteries, e.g. potassium ion batteries. The interaction between potassium ion and 2LG/GPG, as well as the energy barriers for potassium ion migration within and 2LG/GPG were demonstrated in Fig. 4a (detailed in Supplementary Note 12). The 2LG interlayer shows strong interaction (Fig. 4a-I), while the GPG displays moderate interaction (Fig. 4a-II). The visual images for the interactions were demonstrated in Fig. 4a-III and Fig. 4a-IV. The energy barrier for potassium ion migration in 2LG (1.10 eV) is 10 times higher than that in the GPG (0.09 eV), indicating the potassium ion migrate much faster in GPG (Fig. 4a-V and Fig. 4a-VI). The charge-discharge curves for a potassium ion battery at current rate of 1 C and 20 C were illustrated in Fig. 4b, which displayed a capacity of 380 and 330 mAh g$^{-1}$ respectively, indicating a near rate-independent capacity. The capacities of GPG potassium ion batteries at current rate at 0.5, 1.0, 2.0, 5.0, 10, 20 and 50 C are around 400, 380, 370, 350, 340, 330, and 305 mAh g$^{-1}$ respectively (Fig. 4c), with an average CE around 99% (Fig. 4d). To verify the ultra-fast

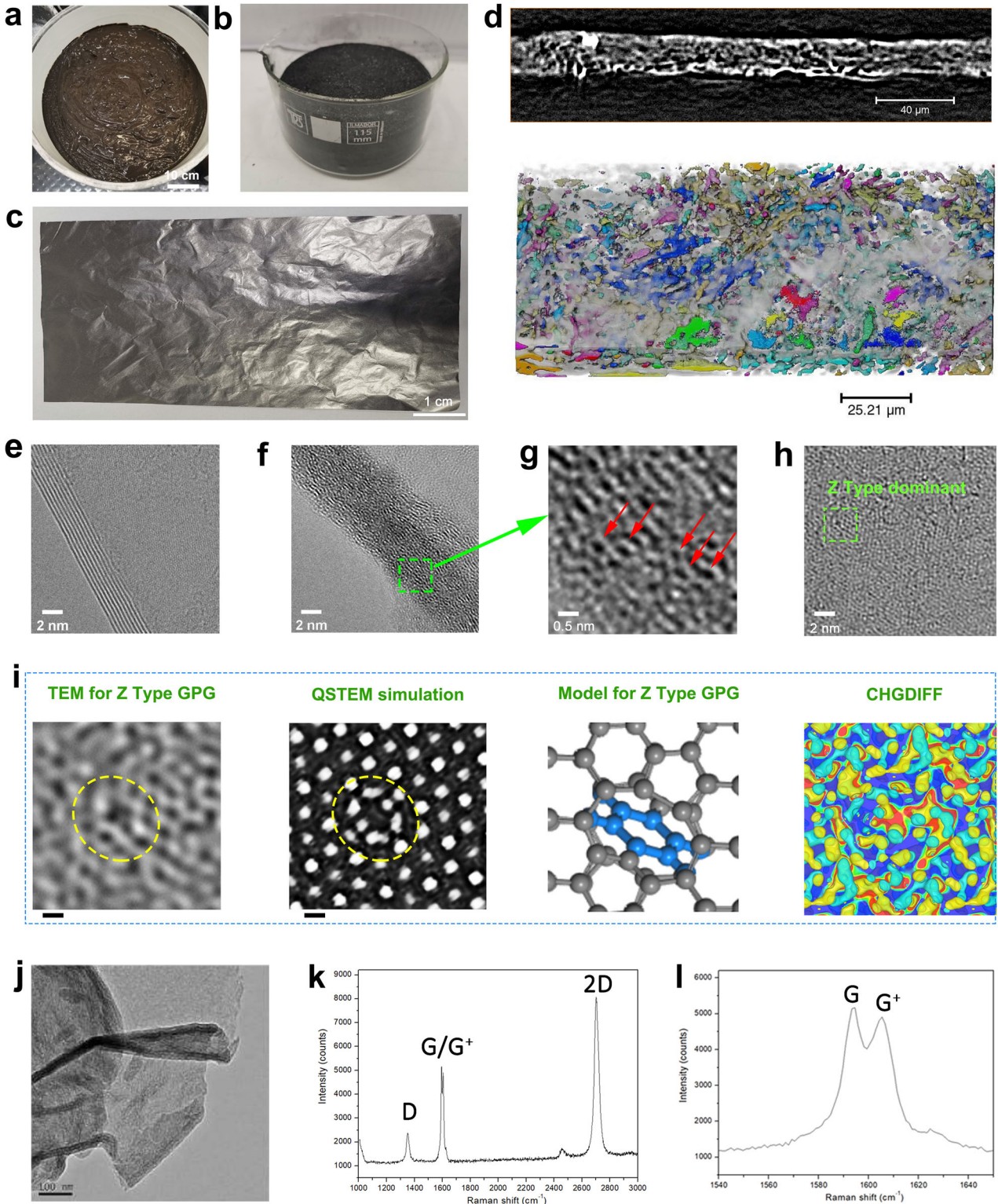

**Fig. 3 | Synthesis and characterizations of Z-type GPG. a** The massive production of GO-P-GO from graphene oxide ink; **b** The GPG formed by pyrolysis of GO-P-GO; **c** The optical image of GPG film; **d** side view of X-CT image for the GPG film and 3D pore-network of the GPG film; **e** TEM image of multi-layer graphene; **f** TEM image, **g** enlarged TEM image, and **h** HRTEM image of Z-type GPG; **i** enlarged HRTEM (first), QSTEM simulation (second), theoretical model (third, grey balls and sticks are the carbon atoms and C-C bonds in graphene plane; blue balls and sticks are the carbon atoms and C-C bonds in the p-phenyl group), and CHGDIFF images (fourth) of Z-type GPG; **j** Typical TEM image, **k** Raman spectrum and **l** corresponding G/G⁺ peaks for Z-type GPG.

charging capability of GPG for PIBs, the RGO and GPG half cells of PIBs were charged within voltage range of 0–3 V at extremely high current rate of around 210 C for 2300 min (≈8000 cycles, Fig. 4e). The RGO PIBs (blue curve in Fig. 4e) shows obvious overpotential

after around 500 cycles and tends to deteriorate afterwards, which is possibly ascribed to the potassium deposition on the electrode surface, and even worse, the formation of potassium dendrites. The GPG PIBs remain stable for ≈8000 cycles at 210 C (pink curve in

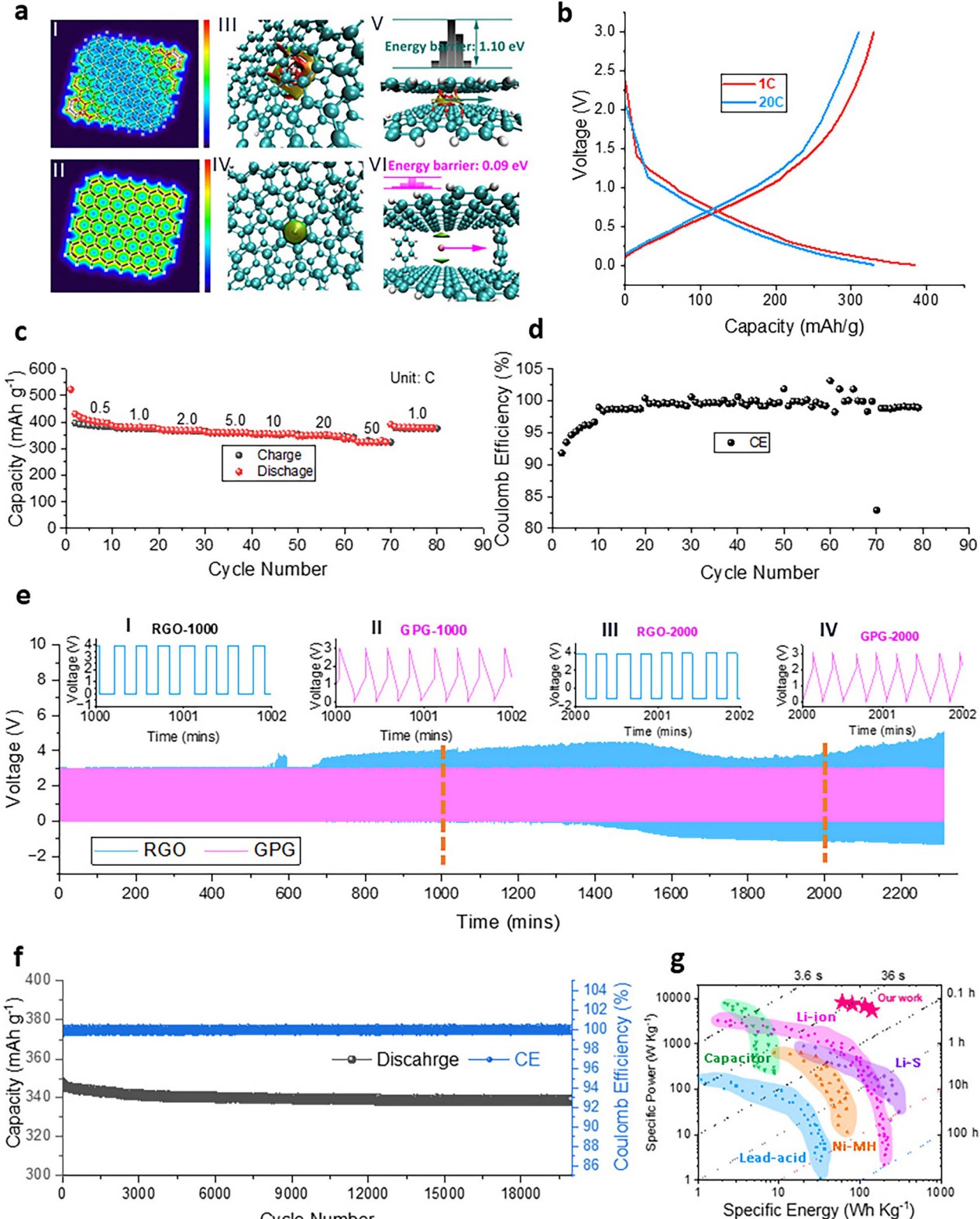

**Fig. 4 | Theoretical and experimental performances of GPG for potassium ion batteries. a** The van der Waals forces within 2LG (I) and GPG (II) and their energy barriers for potassium ion migration (a-I: interaction between graphite layers; a-II: interaction between GPG layers; a-III: visualized interaction between graphite layers with a potassium ion insertion; a-IV: visualized interaction between GPG layers with a potassium ion insertion; a-V: energy barrier for potassium ion migration in graphite layers; a-VI: energy barrier for potassium ion migration in GPG layers; Blue: Attractive force; Green: Neutral; Red: Repulsive force; Blue ball: Carbon atom; White ball: Hydrogen atom); **b** the charge-discharge profiles for potassium ion battery working at 1 C and 20 C; **c** the rate performance for PIBs and **d** the relative CE; **e** the charge-discharge profiles at ultra-high rate of 210 C (1 C = 300 mA g$^{-1}$) for PIBs with RGO and GPG negative electrodes (insert) the enlarged charge-discharge curves at 1000 and 2000 min; **f** the long-term cycle stability and CE of GPG based PIB at 2 C for 20000 cycles (the testing temperature is 25 ± 1 °C and cutoff voltages is 0.01-3.0 V with a safe voltage range from −2.0 to 5.0 V for overpotential protection); **g** the comparison of our work with other battery systems.

Fig. 4e), indicating the potential of the GPG electrode being charged in ≈17.1 s under extreme conditions. When we look on the exaggerated curves for RGO and GPG PIBs at 1000–1002 and 2000–2002 min, the voltage window of RGO PIBs increased rapidly from ≈0 V to ≈4 V and the curves show a square shape (blue insertion curves in Fig. 4e-I and 4e-III), suggesting a huge overpotential and dendrite risk at this high current density[25]. However, the GPG PIBs charge-discharge curve retains a triangle-shape charge-discharge curves, with the overpotential even tending to decrease at 2000–2002 min compared to 1000–1002 min (pink insertion curves in Fig. 4e-II and 4e-IV), this is likely ascribed to the penetration of potassium ions expanding the pathway for migration. The long-term stability of GPG PIBs at current rate of 2 C was shown in Fig. 4f, which demonstrated a reversible capacity of 330 mAh g$^{-1}$ after 20, 000 cycles with a retention rate of 94.3%. Quantitative analysis indicates a mixed charge storage mechanism involving both diffusion and capacitive-controlled processes, as detailed in Supporting Information Fig. S28[26,27]. The specific energy, specific power, and charging rate for the GPG PIBs (based on the mass of electrode) were compared with the widely reported energy storage devices including LIBs, Li-S and supercapacitors, which overwhelms the state-of-the-art existing energy storage device (Fig. 4g).

Meanwhile, the application of GPG for Al| Aluminium Chloride–1-Ethyl-3-Methylimidazolium Chloride (AlCl$_3$-EMIC)|GPG battery was extensively investigated. The ion diffusion coefficient AlCl$_4^-$ ($D_{AlCl4^-}$) in GPG (1.74 × 10$^{-13}$ cm$^2$ s$^{-1}$) is ~10 times larger than that in graphite (1.93 × 10$^{-14}$ cm$^2$ s$^{-1}$) due to the insertion of p-phenyls between the layers of GPG (detailed in Supplementary Note 13). This feature makes GPG be one of the most promising electrode materials for ultra-fast migration of both cations and anions. The migration behaviours of K$^+$ in GPG were also investigated, revealing overwhelming performances for ultra-fast transformation (detailed in Supplementary Note 14–18). As a result, a capacity of 67.1 mAh g$^{-1}$ was achieved at 10 A g$^{-1}$ (fully charged within 24.2 s).

In summary, we propose a new carbon allotrope, GPG nanomaterial, with ultra-high hall mobility through bridging π-π-conjugated p-phenyls between graphene layers. The bridging p-phenyls weakened the van der Waals interlayer forces and strengthened the π electron delocalization of graphene in-plane, expanded the layer spacing and accelerated the ion migration between graphene layers, resulting in an ultra-high electrical conductivity (hall mobility up to 10,000 cm$^2$ V$^{-1}$ s$^{-1}$) and ionic mobility (5.28 × 10$^{-11}$ cm$^2$ s$^{-1}$). Owing to these features, the GPG achieved a capacity of 305 mAh g$^{-1}$ at 50 C, and a maximum current density of 210 C (fully charged within 17.1 s, the best rate and capacity performances ever reported) as K-ion battery negative electrode, and realized the capacity of 78.2 mAh g$^{-1}$ at 10 A g$^{-1}$ (fully charged in 24.2 s) for Al-ion battery positive electrode. This strategy opens new avenues for designing carbon allotropes for ultra-fast rechargeable next-generation batteries. In addition, the bridging of p-phenyls between graphene layers also alters many other physical properties of graphene, e.g., work function, electronic state, fermi level etc., which remains to be further verified.

## Methods
### Materials
Natural graphite powder, sodium cholate, 1-ethyl-3-methylimidazolium chloride, aluminium chloride, poly(methyl methacrylate) and acetone were purchased from Alfa Aesar. The Whatman GF/D type separator was purchased from Sigma-Aldrich. P$_2$O$_5$, K$_2$S$_2$O$_8$, H$_2$SO$_4$ (98 wt.%), NaNO$_2$, HCl solution (37%), NaNO$_3$, p-phenylenediamine, and KMnO$_4$ were purchased from Sinopharm Chemical Reagent Co. All the reagents are Analytical Reagent grade and used without further purification. Accessories related to coin cell and pouch cell assembly (cases, spacers, Cu and Al foils, tabs etc.) were purchased from MSE suppliers.

### Synthesis of GPG precursors
Firstly, graphene oxide (GO) was prepared from natural graphite powder through a modified Hummers' method[28]. Typically, 1 g P$_2$O$_5$ and 1 g K$_2$S$_2$O$_8$ were added to 20 mL of concentrated sulphuric acid (98 wt.%) in ice bath (0 °C), and then natural graphite (2 g) was gradually supplemented. After that, the temperature of the suspension was raised to 80 ± 1 °C and maintained for 6 h under the vigorous stirring. When the mixture cooled to 25 ± 1 °C, 150 mL of deionized (DI) water was then added in an ice water bath. The resulting mixture was filtered and washed several times with DI water and then dried in a vacuum oven at 40 ± 1 °C for a whole night. Finally, the pre-oxidized graphite was obtained. 2 g pre-oxidized graphite and 1 g NaNO$_3$ were added to 92 mL of concentrated sulphuric acid (98 wt.%) in ice bath (0 °C) under stirring. 10 g KMnO$_4$ was gradually added to the suspension with vigorous stirring. The ice bath was then removed, and the temperature of the suspension was raised to 35 °C and maintained for 24 h under the vigorous stirring. 200 mL DI water was added to the mixture followed by addition of 5 mL of H$_2$O$_2$ (5%) to reduce the residual KMnO$_4$ until the colour of the solution turned from dark brown to yellow. The resulting solid graphite oxide was separated and washed with HCl solution to remove metal ions. The resulting mixture was filtered and washed several times with DI water until the pH of the filtrate was neutral and then dried in a vacuum oven at 40 ± 1 °C for 24 h.

Besides, single-layer graphene obtained through liquid phase exfoliation (LPE) and Chemical Vapour Deposition (CVD) were also used as precursors for GPG synthesis. Single-layer graphene was prepared via a liquid phase exfoliation method using high-pressure homogenization. Natural graphite powder (≥99% purity) was first dispersed in an aqueous solution of sodium cholate (5 g L$^{-1}$) as a surfactant. The dispersion was then subjected to high-pressure homogenization using a microfluidizer (Microfluidics M-110P or Homogenizer PSI-40) operated at 2000 ± 20 bar for 100 cycles. The resulting dispersion was centrifuged at 3000 rpm for 6 hours to remove thicker graphite flakes and surfactant residents. The supernatant, enriched in mono- and few-layer graphene, was collected and stored at 4 ± 0.1 °C. The lateral size of graphene flakes typically ranged from 0.5 to 2 μm.

High-quality monolayer graphene was synthesized by low-pressure chemical vapour deposition (CVD) using a copper foil substrate (25 μm thick, ≥99.8% purity, Alfa Aesar). The copper was cleaned in acetic acid to remove surface oxides, rinsed with deionized water, and loaded into a quartz-tube CVD furnace. The furnace was heated to 1,000 °C under a flow of hydrogen gas (100 standard cubic centimetres per minute (sccm)), followed by the introduction of methane (1 sccm) for 30 min to initiate graphene growth. After growth, the furnace was rapidly cooled to room temperature under hydrogen. The resulting graphene film was transferred onto SiO$_2$/Si substrates for further use or onto a precursor solution for subsequent reactions, using a poly(methyl methacrylate)-assisted wet transfer method. The poly(methyl methacrylate) was later removed by acetone immersion, and the graphene was rinsed with isopropanol and dried under nitrogen flow. Raman spectroscopy confirmed monolayer coverage over most of the substrate.

### Synthesis of GPG
The as-prepared GO was uniformly dispersed in aqueous solution, and then p-phenylenediamine, NaNO$_2$ and diluted HCl solution (0.1 M) were sequentially supplemented to bridge p-phenyl between GO through diazotization reaction. The mixture was placed in ice bath (0 °C) and stirred for 24 h, end up to a dark yellow colour, and then centrifuged and washed with ultra-pure water. After drying at 60 ± 1 °C, the GO-P-GO was obtained. Then, the GO-P-GO was reduced to be GPG by hydrothermal reaction at x °C (x = 160, 180 and 200, named as GPG-h-160, GPG-h-180 and GPG-h-200 respectively), or conducted with

high temperature pyrolysis to obtain GPG at T (T = 900, 1200, 1600, 2000, 2400 and 3000 °C, namely GPG-p-900, GPG-p-1200, GPG-p-1600, GPG-p-2000, GPG-p-2400 and GPG-p-3000) under inert atmosphere. Alternatively, single-layer graphene obtained through liquid phase exfoliation (LPE) and Chemical Vapour Deposition (CVD) were also used as precursors to synthesize GPG following a similar approach. It appears the GO precursor ends up in a Z type dominated GPG, while the LPE and CVD graphene precursors predominantly result in H type GPG.

To regulate the functionalization and minimize random p-phenyl insertion, we employed a carefully optimized synthesis protocol above, adjusting the precursor concentration, reaction time, and temperature to control the uniformity and stability of the GPG structure. In particular, the mass ratio of p-phenyl precursor to graphene was controlled at approximately 1:25. Based on the molar mass calculations, this corresponds to an average incorporation of approximately one p-phenyl group per 150 carbon atoms, providing a semi-quantitative assessment of the degree of modification.

Nonetheless, some degree of local variation in p-phenyl group distribution is inevitable, which may contribute to structural heterogeneity. We recognize that achieving complete uniformity is challenging under current synthesis conditions. In our experiments, the GPG samples synthesized from LPE and CVD graphene predominantly exhibited H-type stacking, whereas samples derived from reduced graphene oxide (RGO) tended to favour Z-type stacking. However, in most practical cases, both H- and Z-type configurations coexist randomly within the materials.

## Materials characterizations

The morphology and structure of samples were investigated using scanning electron microscope (SEM, Hitachi S-4800, Japan), transmission electron microscope (HRTEM, TF20/2100 F, FEI) with energy dispersive spectrometer (EDS), and spherical aberration corrected scanning transmission electron microscope (Cs-corrected STEM, JEMARM300F). XPS measurements were carried out with ESCALAB 250, which used monochromatic Al K X-ray source with a pass energy of 30 eV in 0.5 eV step over an area of 650 μm×650 μm. Hall mobility was conducted on Lake Shore 8400. TGA was performed on a Q600 SDT. XRD data were collected on a Rigaku D/max-2200pc with a scan rate of 4° min$^{-1}$. Raman spectra were collected on HORIBA LabRAM HR Evolution with an excitation wavelength of 633 nm. Solid-State NMR structural characterization was conducted with Agilent 600 M, and Fourier Infrared spectroscopy were obtained with Bruker Vertex 70 FTIR spectrophotometer. Imaging Processing, Data Analysis, and Image-Based Modelling: micro-CT projections were reconstructed using a filtered-back projection algorithm (XMReconstructor, Carl Zeiss Inc.). The reconstructed micro-CT datasets were imported into Avizo 2023.2 (ThermoFisher) for further segmentation and visualization. A non-local means filter was applied to increase the signal to noise ratio. The pristine material, imaged by micro-CT, were segmented by ilastik software into two phases consisting of pores and carbon film. The RVE Analysis: The sub volume is extracted from the original material, and the microstructural parameters (such as tortuosity) of the pore phase were then determined by using Matlab R2021b with the TauFactor plugin[29]. The RVEs were implemented to determine the relationship between the fraction of total volume, volume fraction and tortuosity (τlocal) on X, Y and Z directions. The local effective diffusivity coefficient was also determined on three directions. Moreover, the TauFactor software can provide the fluxes for the electrodes along the X, Y and Z-axis direction.

## Theoretical calculations

Computer modelling was performed using the Vienna Ab initio Simulation Package (VASP) with supplied projector augmented wave potentials for core electrons[30]. The generalized gradient approximation of Perdew-Becke-Ern-zerhof was used for the exchange correlation function[31]. The conjugate gradient algorithm was used in the structural optimization of GPG, providing a convergence of $10^{-5}$ eV in total energy. The cutoff energy was set to 500 eV with a 5 × 5 × 1 K-point mesh to represent the Brillouin zone. The transition state and diffusion pathway of K$^+$ and AlCl$_4^-$ ions were calculated using the climbing image nudged elastic band (CI-NEB) method.

The finite structure of GPG saturated by hydrogen atoms was also optimized using CP2K[32] with the help of Multiwfn[33] to verify the accuracy of models. PBE functional with TZVP-MOLOPT basis set and D3BJ dispersion correction was adopted with a 5×5×1 K-point grid. The cutoff and relative cutoff energy were set to 800 Ry and 60 Ry. The convergence limits for step size and RMS step size were $3 × 10^{-3}$ and $1.5 × 10^{-3}$ and the convergence limits for max force and RMS force were $4.5 × 10^{-4}$ and $3 × 10^{-4}$. The pressure tolerance was set to 100 Bar.

First-principles finite temperature molecular dynamics (MD) simulations were used to check the stability of the structures in 1 × 3 × 3 supercells for 50,000 fs with a time step of 1 fs. Large-scale molecular dynamic simulations were performed in a 5 × 26 × 15 supercell with 31200 carbon atoms using the large-scale atomic/molecular massively parallel simulator[34] and MedeA environment[34]. The dynamic process was performed with the NVT ensemble and Nosé-Hoover thermostat at a temperature of 300-3300 K. The number of steps is 100000 for NVT-MD simulation and the time step is 2 fs, resulting in the total simulation time of 1 ns. (Supplementary Figs. S20 and S21, and Supplementary movie 1 (**Z-type GPG @1200 K**), Supplementary movie 2 (**Z-type GPG @1900 K**), Supplementary movie 3 (**Z-type GPG @2600 K**), Supplementary movie 4 (**Z-type GPG @3300 K**), Supplementary movie 5 (**H-type GPG @1200 K**), Supplementary movie 6 (**H-type GPG @1900 K**), Supplementary movie 7 (**H-type GPG @2600 K**), Supplementary movie 8 (**H-type GPG @3300 K**).) The strongly constrained and appropriately normed (SCAN) meta-GGA with the revised Vydrovvan Voorhis no-local correlation functional (SCAN +rVV10) is utilized for the dispersion correction[35]. This approach yields superior predictions of both energetic and structural properties for many types of bonding, addressing limitations that many non-empirical semi-local functionals struggle to encompass[36–38].

## Electrochemical measurements

2032 coin-type cells or soft packing batteries were assembled to investigate the performances of active materials for K$^+$ and AlCl$_4^-$ ions migration. In K-ion batteries, potassium foil was used as counter electrode, and the separator was a Whatman GF/D type (47 mm diameter, Thickness: ~675 μm, Lateral Dimension: 4.7 cm, Porosity: 70–85%, Average Pore Size: ~2.7 μm). The electrolyte was 4.0 M potassium bis(fluorosulfonyl)imide (KFSI) in a diethylene glycol dimethyl ether (DEGDME) solvent. As for Al| Aluminium Chloride−1-Ethyl-3-Methylimidazolium Chloride (AlCl$_3$-EMIC) | GPG batteries, aluminium foil was used as counter electrode, polypropylene membrane from Celgard was used as separator, and AlCl$_3$-EMIC was used as electrolyte. To validate the reproducibility of the electrochemical performances, all the loading of active material GPG on the current collector is around 2 mg cm$^{-2}$, which falls within the common range for laboratory-scale coin cell assembly and performance evaluation.

For the potassium-ion battery, 2032-type coin cells were assembled in an argon-filled glovebox (O$_2$ and H$_2$O levels <0.1 ppm). The working electrodes were prepared using a composite of active material, conductive carbon (Super P), and PVDF binder in a weight ratio of 80:10:10. N-methyl-2-pyrrolidone (NMP, ≥99.5%, Sigma-Aldrich) was used as the solvent. The slurry was cast onto copper foil (10 μm thick, ≥99.9% purity) using a doctor blade with a 150 μm gap, followed by drying under vacuum at 80 °C for 12 hours. The dried electrodes were then punched into 12 mm discs using a manual die cutter. The

Poly(vinylidene fluoride) binder (PVDF) used (Kynar HSV900, Arkema) had a molecular weight of approximately 534,000 g/mol. The copper foil was used without further surface treatment. The counter/reference electrode was potassium metal (Disk: $15 \pm 0.1$ mm for diameter and $200 \pm 10$ μm for thickness), and a glass fibre separator (Whatman GF/D, diameter 19 mm) was employed. Approximately $80 \pm 1$ μL of electrolyte was used per coin cell. Stainless steel cases and springs (2032 standard) were used, and each cell included a single-sided coated electrode. For the aluminium-ion battery, a 1 cm × 2 cm soft package-type pouch cell was assembled. The electrodes were prepared similarly by casting the slurry onto aluminium foil (15 μm thick, ≥99.9% purity), followed by vacuum drying and precision cutting using a blade cutter. The pouch cell was fabricated in a dry room, and external pressure of approximately 1 MPa was applied during cycling using a clamp fixture. Prior to sealing, the electrolyte (1-ethyl-3-methylimidazolium chloride-aluminium chloride, EMIMCl–AlCl$_3$, in a molar ratio of 1:1.3) was injected manually, followed by resting and multiple vacuum degassing cycles to remove trapped gases. The aluminium foil current collector was used as received, without etching or modification. The electrolyte volume was $200 \pm 10$ μL per pouch cell. Electrodes were single side coated with an areal loading of $2 \pm 0.1$ mg/cm$^2$.

The Al|AlCl$_3$-EMIC | GPG soft packing batteries were assembled and tested using Ti or Ta foil as current collectors. The rate performances, galvanostatic charge/discharge cycling profiles and the cycling performances were carried out using a LAND-CT2001A battery test system (Shanghai, China) within the voltage window of 0.01-3.0 V for K-ion batteries and 0.01-2.5 V for Al|AlCl$_3$-EMIC | GPG batteries. The cyclic voltammograms (CVs) tests, two-electrode system tests were carried out on a CHI660e electrochemical workstation. Electrochemical impedance spectroscopy (EIS) measurements were carried out in potentiostatic mode with a sinusoidal excitation amplitude of 5 mV (rms). The frequency range spanned from 1 kHz to 10 mHz, with 10 data points per decade to ensure sufficient resolution across the spectrum. Prior to each EIS measurement, the cell was held at open-circuit voltage (OCV) for 1 h to allow for equilibration and ensure quasi-stationary conditions. The measurements were conducted using a computer-controlled electrochemical workstation (Bio-Logic BCS810). All measurements are under ambient atmosphere with a temperature of $25 \pm 1$ °C.

## Data availability

The authors declare that all the relevant data are available within the paper and its Supplementary Information file. Additional supporting data of this study are available from the corresponding author on request. Source data are provided as a Source Data File and Supplementary Data 1 are provided with this paper. Source data are provided with this paper.

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

## Acknowledgements

This work was supported by the Postdoctoral Fellowship Programme (Grant No. PC2022020). H.L., J.B.R. and P.R.S. would like to acknowledge the Faraday Institution's LiSTAR research programme (FIRG0058). H.L. and J.B.R. would like to acknowledge InnovateUK for their funding of the QUASSLIS project (576353). P.R.S. also acknowledges the Royal Academy of Engineering Chair in Emerging Technologies (CiET1718\59). H.L. and H.C. thank the use of the University of Oxford Advanced Research Computing (ARC) facility for resources in carrying out this work (10.5281/zenodo.22558). B.L. acknowledges the National Natural Science Foundation of China (No. 52202154), Chinese Universities Scientific Fund (No. 15055002), Beijing Natural Science Foundation (No. 2252039), and the High-performance Computing Platform of China Agricultural University, as well as the support from the National Supercomputer Centre in Tianjin and Tianhe new generation supercomputer. Prof. Richard G. Compton in the University of Oxford is appreciated for the great support during this programme.

## Author contributions

H.L. conceived and designed the idea and co-wrote the paper and supervised the project and is responsible for the infrastructure and project direction. H. L., H. C. and B. P. discussed the idea, experimentally realized the study, analysed the data and co-wrote the paper, and these works were assisted by H.C., B.P., J.Z., S.R.P.S., Y.C.W., S.Z. and P.R.S. The overall project was led and managed by H.L., B.L., K.S.N. and J.B.R. All the authors discussed the results, commented on and revised the manuscript.

## Competing interests

The authors declare no competing interests.
