## [Peer Review File · Nature Communications]

Van-der-Waals-forces-modulated Graphene-P-phenyl-Graphene Carbon Allotropes

Corresponding Author: Dr James Robinson

Version 0:

Reviewer comments:

Reviewer #1

(Remarks to the Author)

The authors propose a new carbon allotrope Graphene-P-phenyl-Graphene (GPG) by DFT and MD methods, and they also synthesize GPG by inserting π - π -conjugated p-phenyls into the layers of graphene and connecting the graphene layers by C-C σ bonds. The chosen bridging p-phenyls could weaken the van der Waals interlayer forces, strengthen the π electron delocalization of graphene in-plane, expand the layer spacing and accelerate the ion migration between graphene layers. The authors also perform potassium ion batteries assembled with GPG, exhibiting high reversible capacity, long-life stability, and excellent charge-discharge rate. Their methods are reasonable, and the results are interesting. However, major revisions are needed to improve the work. Some comments are listed below:

1.How to confirm that the actual material structure obtained by experiment is consistent with the theoretical model structure is a key challenge in materials science research. In my opinion, the prediction part including Figure 1 is mainly used to introduce GPG. But the model is not so consistent of the real experimental material. Firstly, the real materials have defects. Secondly, three or more layers can have p-phenyls between them. The authors need to study some more models with defects or more layers theoretically to confirm their data.

2.Why choose K-ion batteries instead of Na or Li-ion batteries? Li-ion batteries with GPG may provide better performance, due to the large layer spacing and good ion migration.

3.The model optimization section only discusses the effect of interlayer spacing, but ignores the potential effect of different rotation angles of p-phenyl. It is recommended to further explore the stability differences between Z-type and H-type structures with different rotation angles of p-phenyl.

4.In addition, to confirm the stability, the phonon dispersion of GPG is not provided.

5.Regarding the large discrepancy between the order of change of average carbon energy (AEPC) and free energy (FE) in Figure S3, a more in-depth discussion is recommended. In particular, the low AEPC but very high FE of the C0 structure in Figure S3 may be due to its special electron localization or bond length stress distribution. This phenomenon should be explained by more theoretical analysis or supplementary calculations.

6.The manuscript mentions molecular modifications such as vinyl, hydroxyethyl, amide, p-phenyl and biphenyl, but lacks comparative data on their specific effects. It is recommended to supplement comparative data on the charge transfer ability, electronic properties and interlayer adjustment ability of these molecules to clarify the reason for choosing p-phenyl. The effects of these molecular modifications on the performance of GPG should be demonstrated through experimental data or calculation results to prove the rationality of the choice of p-phenyl.

7.The pictures of ball-and-stick models in Figure 2b-2c and Figure 3c are not needed. In my opinion, Figure 1 has provided good model images.

8.Redundancy of charge distribution difference maps. Figure 2d shows the three-dimensional charge distribution difference of GPG, but the manuscript shows 6 charge distribution maps repeatedly (Figure 2d-i). Just keep the key images. The charge transfer of different structures should be compared through consistent settings of isosurfaces. In addition, specific

charge transfer values should be provided to quantitatively analyze how charge redistribution affects the electronic properties of the material, especially the specific effects of different molecular modifications on charge distribution.

9. In this paper, the high-resolution transmission electron microscopy (HRTEM) image and transmission electron microscopy (TEM) image of GPG prepared by the method of introducing p-phenyl groups both show that the structure is rough and defective. It is also possible to randomly introduce p-phenyl groups between graphene layers during the experiments. How many p-phenyl groups in GPG samples? These factors can affect the final structure and properties of GPG. Please further elaborate on these issues to clarify how to effectively control the uniformity and stability of the structure during the preparation of GPG.

Some typos in the manuscripts:

1) 'dynastically' in the sentence 'The carbon allotrope of GPG is thermally and dynastically stable'

2) Figure 2c should be checked in the sentence 'The band structures and DOS of both Z and H type GPG were demonstrated in Figure 2c'.

Reviewer #2

(Remarks to the Author)

In this manuscript, the author reports a new allotrope to the nanocarbon family, Graphene-P-phenyl-Graphene (GPG), synthesized by inserting π - π -conjugated p-phenyls into the layers of graphene and connecting the graphene layers by C-C σ bonds. My overall view of the paper is that while the work is of potential interest for publication, there are at least following critical weaknesses that need to be addressed before the work can be considered for publication in Nature Communications. Here are detailed reasons and comments.

1. The author needs to give the information of the area capacity when conducting the battery performance test. It is worth noting that too low an area capacity is of little significance for studying the performance improvement of K-ion battery. It is recommended to explain in this article.
2. In order to better illustrate the charging and discharging states of GPG cycled at the voltage, it is necessary to add the in situ XRD measurements of the sample at 0–3.0 V in this work.
3. This work requires the addition of typical Nyquist plots of the battery at various SOC values, which can analyze the stability under the voltage charging conditions.
4. The article mentions that graphene obtained from CVD preparation can also synthesize GPG. It is suggested to provide the electrochemical properties of it. Please explain the reason for the difference in electrochemical properties of GPG obtained from different graphene.

Reviewer #3

(Remarks to the Author)

This work reported a new type of allotrope by inserting π - π -conjugated p-phenyls into the layers of graphene. Owing to the expanded interlayer spacing, the reported allotrope exhibits high hall mobility and excellent rate performance as well as cycling stability when used as K ion battery anode. However, following issues on the reported allotrope remains to be discussed before publication.

- (1) In the section of experimental verification, the author characterized interlayer spacing and chemical composition of prepared allotrope GPG via AFM, TEM, Raman, FTIR in combination with NMR. Can the author give more direct experimental evidence on the specific configuration of GPG, especially the bonding between P-phenyl and graphene. Will characterizations such as aberration-corrected transmission electron microscopy and noncontact atomic force microscopy (which has been used in Science 2021, 372, 852-856 and Nature 2023, 614, 95-101) be helpful for structural verification?
- (2) Did the author consider dispersion correction in the calculations of formation energy? If did, corresponding descriptions should be supplemented in section 1.5 of supplementary materials.
- (3) The author screened the suitable bridge molecules only by comparing the formation energies of different configurations. However, it can be seen that the formation of configuration A2, B2 and C2 in Figure S3 needs to experience chemical reaction processes such as removing NH₂ group. What about the kinetic energy barrier during these processes? In this regard, the energy barrier during GPG formation is also an important indicator which should be considered, especially for configurations with close formation energy (such as configuration B2 and C2).
- (4) As to the scalable synthesis of GPG, is the bonding between P-phenyl and graphene same as that in GPG prepared from graphene. It is known that the carbon atom in P-phenyl can also bond with the oxygen atom besides carbon atom in graphene oxide. In this regard, precise structure of GPG produced from scalable synthesis process should be verified.
- (5) For the electrochemical performance evaluation (Page 12) as K ion battery, contributions from diffusion and capacitive controlled behavior are suggested to be analyzed (such as Figure 6 in Adv. Energy Mater. 2023, 13, 2300442 and Figure 3 in Adv. Energy Mater. 2021, 11, 2002981).
- (6) According to the cycling profile comparison shown in Table S2, the listed studies are all the references published before 2017. Considering the timeliness of the study, recent published work should be supplemented in the comparison.

Version 1:

Reviewer comments:

Reviewer #1

(Remarks to the Author)

The authors have responded all my comments. I think the manuscript could be published.

One typo mentioned in previous comments (comment-10 makred by the authors) 'dynastically'should be 'dyanmically'.

Reviewer #2

(Remarks to the Author)

According to the referee's comments, the authors have corrected the contents of the paper and have answered the referee's problem in detail in the revised manuscript. I believe that this revised manuscript is suitable for publication in Nature Communications.

Reviewer #3

(Remarks to the Author)

The revised manuscript has addressed all the reviewers' comments and is now acceptable.

Response to Reviewers

Response to Reviewer#1

Comment–0:

The authors propose a new carbon allotrope Graphene-P-phenyl-Graphene (GPG) by DFT and MD methods, and they also synthesize GPG by inserting π - π -conjugated p-phenyls into the layers of graphene and connecting the graphene layers by C-C σ bonds. The chosen bridging p-phenyls could weaken the van der Waals interlayer forces, strengthen the π electron delocalization of graphene in-plane, expand the layer spacing and accelerate the ion migration between graphene layers. The authors also perform potassium ion batteries assembled with GPG, exhibiting high reversible capacity, long-life stability, and excellent charge-discharge rate. Their methods are reasonable, and the results are interesting. However, major revisions are needed to improve the work. Some comments are listed below:

Response to Comments–0: Thank you for your positive comments. We sincerely appreciate your thoughtful and encouraging feedback on our manuscript. In response to your enumerated comments, we have supplemented data from both experimental and theoretical sides and made the requested revisions and believe that the additional information provided significantly strengthens the quality of our manuscript. We are looking forward to your approval, thanks.

Comment–1:

1. How to confirm that the actual material structure obtained by experiment is consistent with the theoretical model structure is a key challenge in materials science research. In my opinion, the prediction part including Figure 1 is mainly used to introduce GPG. But the model is not so consistent of the real experimental material. Firstly, the real materials have defects. Secondly, three or more layers can have p-phenyls between them. The authors need to study some more models with defects or more layers theoretically to confirm their data.

Response to Comments–1: Thanks for your comments. We thank the reviewer for raising this critical and insightful point. Indeed, accurately correlating theoretical model structures with real experimental materials remains a key challenge in materials science, especially given the inevitable presence of defects and multilayer configurations in practical samples.

We acknowledge that the idealized model presented in Figure 1 serves primarily to introduce and highlight the fundamental structural motifs of GPG. While the model may not fully capture all complexities of the experimentally synthesized material—such as structural defects and the possible presence of p-phenyl groups between three or more graphene layers—it reflects the dominant and representative configurations that are most likely to govern the material’s properties.

To bridge the gap between theory and experiment, we have compared the theoretically predicted H- and Z-type structures with HRTEM images acquired along the same crystallographic directions. These comparisons show a notable degree of agreement, particularly in atomic alignment and lattice symmetry, supporting the relevance of our models. We appreciate the reviewer’s suggestion, which has helped us clarify the relationship between our theoretical predictions and experimental observations in the revised manuscript.

The relative content for the H-type GPG has been supplemented and is detailed as follows:

Figure R1: (g) TEM image of GPG; (h) Enlarged TEM and HRTEM (inserted) images of GPG; (i) HRTEM image and modelling structure of the top view for H-type GPG; (j) Side view of the modelling structure of H-type GPG (the ball-type atoms are visible and highlighted in the top view).

*“Following the TEM image of GPG synthesized from CVD graphene (**Figure 2g**), the enlarged HRTEM image (**Figure 2h**) provides a clearer view of the atomic structure of H-type GPG. The region highlighted by the red square is further magnified in the inset, revealing a well-defined hexagonal atomic lattice. This lattice matches precisely with the top-view structural model of H-type GPG (**Figure 2i**), where the atomic positions exhibit excellent alignment. The corresponding side-view model (**Figure 2j**) confirms that the regular hexagonal lattice arises from the top graphene layer (highlighted in pink), while the atoms observed at the centre and periphery of the structure (in lilac) originate from the upper shoulder atoms of the *p*-phenyl groups.”*

The Z-type configuration dominates the GPG synthesized from oxidized graphene oxide, where structural defects are typically present. The HRTEM top-view image of the Z-type GPG (**Figure R1a**) reveals that while the hexagonal ring structure of the graphene plane is partially retained, it becomes more disordered. A comparison between the defect locations (**Figure R1b**) and the Z-type GPG model (**Figure R1c**) shows a good alignment of atomic positions (**Figure R1d**). Additionally, the QSTEM simulation closely matches the HRTEM image (**Figure R1e, f**). The side view of the Z-type GPG (**Figure R1g**) highlights the top-layer atoms characteristic of this configuration.

Figure R2: (a) TEM image of GPG synthesized from graphene oxide precursors; (b) Enlarged HRTEM image and (c) corresponding top-view structural model of Z-type GPG; (d) Overlay of the HRTEM image and the top-view model showing atomic alignment in Z-type GPG; (e) Region of the HRTEM image corresponding to (f) QSTEM simulation; (g) Side-view model of Z-type GPG, with top-layer atoms highlighted (represented as ball-type atoms in the top view). Scale bar: 0.5 Å.

“The HRTEM image on the top view of GPG shows a plane defect derived from reconstruction and deformation of carbon atoms due to the bonding of p-phenyl. The QSTEM simulation well-match the HRTEM image, which was obtained from quantitative simulation of the Z-type GPG model.”

Moreover, we agree that further theoretical exploration of defective and multilayered models would provide deeper insight. We are extending our simulations to include these more complex structures. As rightly pointed out, actual materials inevitably contain defects and variations such as multilayer configurations, which are difficult to capture fully in idealized models. In our initial study, the models shown in Figure 1 were designed to present the foundational structural motifs of GPG, serving as a starting point to understand the material’s underlying architecture. However, to move beyond these idealized representations and more closely mirror experimental realities, we have now extended our theoretical work to include additional models that account for common types of structural imperfections.

We began by examining the nature of defects already inherent in the previously presented structures. Interestingly, the original Z-type GPG model contains a built-in point defect, while the H-type model was constructed as a pristine reference. Building upon this, we deliberately introduced new defects into both structures. In the case of the H-type GPG, we removed a single carbon atom from the graphene plane (now referred to as GPG-V1, shown in **Figure R3a**), and for the Z-type GPG, we introduced two additional carbon vacancies (GPG-V2, shown in **Figure R3b**).

Upon relaxation through DFT optimization, we found that the defected H-type structure (GPG-V1) spontaneously transformed into a configuration nearly identical to the original Z-type GPG, suggesting a structural convergence between these forms under certain defect conditions. The more heavily defected GPG-V2 exhibited greater disorder, representing a more realistic and complex form of GPG.

These findings allowed us to reinterpret our models in the context of experimental material diversity. The H-type GPG can now be understood as the idealized, non-defective structure; GPG-V1 (and by extension, Z-type GPG) reflects a single-point-defect configuration; and GPG-V2 captures a more disordered, defect-rich variant. We believe that these structures likely co-exist within real samples and collectively contribute to the observed properties.

Importantly, our Raman spectroscopy data also supports this interpretation, as the appearance of a D peak confirms the presence of defects in the synthesized GPG material. To ensure transparency and reproducibility, we have included all relevant **CIF** files in the updated **Source Data** package.

This expanded modelling effort helps bridge the gap between theory and experiment and adds greater nuance to our understanding of GPG's structure–property relationships. We are grateful to the reviewer for encouraging this deeper investigation.

Figure R3: (a) The GPG-V1 model with single defect; (b) CHGDIFF image of GPG-V1 model; (c) GPG-V2 model with dual defects; (d) CHGDIFF image of GPG-V2 model

For the multiple layered GPG, we have extended our theoretical investigation to include additional models representing multilayer configurations. Specifically, we constructed a four-layer H-type GPG and a three-layer Z-type GPG (**Figure R4**). These multilayer models exhibit structural characteristics that are highly consistent with their bilayer counterparts. In both the H-type and Z-type forms, the interlayer arrangements and key structural motifs remain preserved, indicating that the dominant features of the GPG architecture are maintained even with increased stacking.

We would like to clarify that for GPG synthesized via CVD graphene—typically in film form—we can reliably obtain double-layer GPG structures, which are the basis of the models presented in the main text. However, in the case of GPG derived from reduced graphene oxide (RGO) or exfoliated graphene, which are more commonly obtained as powders, multilayer stacking is more prevalent. Nevertheless, our comparative analysis suggests that these multilayer structures are essentially extensions of the

bilayer forms along the c-axis, and the overall atomic arrangement remains fundamentally similar. This continuity supports the validity of using double-layer models as representative building blocks for understanding both the local and extended structural features of GPG materials.

We hope this expanded modelling and clarification addresses the reviewer's concern and helps to further align the theoretical framework with experimental conditions.

Figure R4: (a) The 3-layer Z-type GPG model; (b) CHGDIF image of 3-layer Z-type GPG model; (c) The 4-layer H-type GPG model; (d) CHGDIF image of 4-layer H-type GPG model.

The H- and Z-type GPG models presented in our study represent the most stable and dominant structural motifs observed in the synthesized GPG materials. While these models capture the essential atomic arrangements, we fully acknowledge that a variety of structural forms—including those with defects and multilayer stacking—are likely to coexist in real materials.

Recognizing this complexity, we have expanded our theoretical work to include models incorporating both structural defects and multilayer configurations. These new models are now discussed in detail in the revised manuscript (see **Figure S15** and **S16**). This expanded analysis provides a more comprehensive picture of the possible structural diversity within GPG materials and supports the relevance of our initial models as foundational representations.

We also agree with the reviewer that further systematic investigation of a broader range of structural variants will be crucial, particularly in relation to tailoring the material for specific applications. We are actively pursuing this direction in ongoing work, using more controlled synthesis and modeling approaches to explore how structural variations impact electronic, optical, and catalytic properties.

To reflect these updates, we have added the following statement to the main text:

“In addition to the idealized H- and Z-type models, we have explored additional configurations incorporating point defects and multilayer stacking, which likely coexist in real materials. These expanded models further confirm the structural stability and representative nature of the H- and Z-type motifs while highlighting the complexity and tunability of GPG materials.”

Comment–2:

Why choose K-ion batteries instead of Na or Li-ion batteries? Li-ion batteries with GPG may provide better performance, due to the large layer spacing and good ion migration.

Response to Comments–2: Thank you for your insightful comment. We selected K-ion batteries for this study primarily due to their advantages in terms of cost-effectiveness, natural abundance of potassium resources, and their strong potential for large-scale energy storage applications. Although Li-ion batteries typically exhibit higher energy densities and performance in certain aspects, K-ion batteries benefit from a larger ionic radius and more favourable interlayer diffusion dynamics, which synergize well with the properties of GPG. While Na-ion batteries were also considered, the compatibility between graphitic carbon structures and sodium ions is normally quite poor, often resulting in limited electrochemical performance. Specifically, the low affinity and sluggish kinetics of Na⁺ within graphitic layers pose significant challenges.

To further address your suggestion, we have now included a comparative evaluation of GPG in Li- and Na-ion battery systems (**Figure R5**). The results show that although the initial discharge capacities of Na- and Li-ion batteries with GPG electrodes are reasonably high (520 mAh g⁻¹ for Na-ion and 1200 mAh g⁻¹ for Li-ion in the first cycle), their reversible capacities drop substantially in the second cycle (only 157 mAh g⁻¹ for Na-ion and 569 mAh g⁻¹ for Li-ion). This performance degradation likely stems from the fact that, while the large interlayer spacing of GPG facilitates initial ion adsorption on the surface and trapping in defects, it does not support stable, reversible intercalation of Na⁺ and Li⁺ ions. This may also lead to the formation of inactive deposits of "dead" Na and Li.

In contrast, GPG electrodes in potassium-ion batteries demonstrate markedly more stable and reversible electrochemical behaviour, characterized by higher initial coulombic efficiency and superior long-term capacity retention. This enhanced performance is likely attributed to the precisely engineered interlayer architecture, which offers a more favourable environment for potassium-ion storage compared to sodium- or lithium-ion systems. These findings underscore the intrinsic compatibility of

GPG materials with potassium-ion electrochemistry and highlight their potential for high-performance energy storage applications.

Figure R5: the charge-discharge profile of GPG as anode electrodes for (a) K-ion batteries; (b) Na-ion batteries and (c) Li-ion batteries

Accordingly, we have expanded the discussion in the **Supporting Information Figure S25** and updated with the following statement:

“We tested the electrochemical performance of GPG electrodes in Li-, Na-, and K-ion batteries. Among them, K-ion batteries exhibited the most promising results in terms of initial coulombic efficiency, reversible capacity, and long-term cycling stability.”

Comment–3:

*The model optimization section only discusses the effect of interlayer spacing but ignores the potential effect of different rotation angles of *p*-phenyl. It is recommended to further explore the stability differences between Z-type and H-type structures with different rotation angles of *p*-phenyl.*

Response to Comments–3: Thanks for your comments and insightful suggestion. We agree that the rotation angles of the *p*-phenyl groups can significantly influence the stability of both Z-type and H-type structures and warrant further investigation.

In our study, we initially constructed both Z-type and H-type GPG structures with randomly assigned rotation angles for the *p*-phenyl groups. During the structure optimization process—encompassing both atomic positions and electronic states—the systems consistently converged to specific rotation angles, suggesting that these represent the most thermodynamically favourable conformations.

To further examine the influence of rotation angle, for instance, we conducted a detailed investigation on the Z-type GPG structure by varying the rotation and intersection angles between the *p*-phenyl group and the graphene plane (40°, 50°, 60°, and 70°), as illustrated in **Figure R6**. Despite these variations, all structures converged upon optimization to a configuration with an intersection angle of approximately 54.3°, reaffirming that this conformation represents the lowest energy state.

Figure R6: the structure optimization of Z-type GPG with varying rotation angles and intersection angles between the p-phenyl group and the graphene horizon (A: 40° , B: 50° , C: 60° , D: 70°)

A similar behaviour was observed for H-type GPG structures. Regardless of the initial rotation angle, all models relaxed to the same final conformation upon optimization, as shown in **Figure R7**. This is consistent with the expectation that the minimized steric hindrance in the optimized H-type configuration contributes to its energetic stability.

Figure R7: Optimized H-type GPG structures: (a) side view; (b) top view

Our results indicate that moderate variations in the rotation angles of p-phenyl linkers have negligible effects on the overall thermodynamic stability. However, substantial deviations can introduce structural strain and alter the electronic properties, emphasizing the robustness of the optimized conformations.

All relevant CIF files supporting this analysis have been included in the **Source Data**. Additionally, the following text has been added to the main manuscript:

“Additional DFT calculations were carried out using initial configurations with varying rotation and intersection angles (40°, 50°, 60°, and 70°) between the p-phenyl groups and the graphene plane, confirming the thermodynamic preference for the formation of Z-type and H-type GPG structures.”

Comment–4: In addition, to confirm the stability, the phonon dispersion of GPG is not provided.

Response to Comments–4: Thanks for your comments. It is very professional since the phonon dispersion is vital to understand the stability of GPG. To further confirm the dynamical stability of GPG, we have conducted phonon dispersion calculations.

The phonon dispersion and DOS are calculated using the General Utility Lattice Program (GULP)^{R1}, within Tersoff forcefield^{R2, R3}.

References

- R1** Gale, J. D. & Rohl, A. L. The General Utility Lattice Program (GULP). *Molecular Simulation* **29**, 291-341 (2003). <https://doi.org/10.1080/0892702031000104887>
- R2** Tersoff, J. Empirical Interatomic Potential for Carbon, with Applications to Amorphous Carbon. *Physical Review Letters* **61**, 2879-2882 (1988). <https://doi.org/10.1103/PhysRevLett.61.2879>
- R3** de Brito Mota, F., Justo, J. F. & Fazzio, A. Hydrogen role on the properties of amorphous

silicon nitride. *Journal of Applied Physics* **86**, 1843-1847 (1999).

<https://doi.org/10.1063/1.370977>

To gain insight into the vibrational properties of both H-type and Z-type GPG structures, we calculated the full phonon dispersion relations and phonon density of states (DOS), as presented in **Figure R8**. Panels **a–c** shows the results for the H-type configuration, while panels **d–f** depicts the corresponding data for the Z-type.

The two-dimensional phonon dispersion of H-type GPG with frequency from 0 to 2500 cm^{-1} is shown in **Figure R8a**, which is massively complex compared to graphene.^{R4} The phonon dispersion of the H-type GPG exhibits well-separated acoustic and optical branches. The structural distortions in the H-type GPG due to introducing sp^3 carbon atoms break the inherent symmetry of the graphene plane in a certain extent, which may allow phonon modes that were previously forbidden, or alter the frequencies of existing modes for graphene. That is why the phonon dispersion of the H-type GPG is getting more complicated than pure graphene. Notably, the phonon DOS of the H-type GPG (**Figure R8b**) shows multiple peaks corresponding to low-frequency acoustic modes (200–1000 cm^{-1}) and high-frequency optical modes (1400–2500 cm^{-1}). A closer examination of this frequency window from 1400–1750 cm^{-1} (**Figure R8c**) reveals distinct phonon modes, including the characteristic mode for graphene G peak around 1600 cm^{-1} , indicative of in-plane vibrations of sp^2 carbon networks.^{R4}

Similarly, the Z-type GPG structure displays a dense phonon spectrum (**Figure R8d**), with a comparable maximum frequency but slight variations in mode distribution due to its different geometry. One of the main variations between H and Z-type GPG aroused by the defects in Z-type GPG which introduced the localized modes that introduced the frequencies not originally present in the phonon spectrum which is associated with local vibrations related to the defects. Also, defects can act as centres for phonon scattering. This scattering affects the phonon mean free path, thereby influencing

the material's thermal conductivity and other thermodynamic properties. Near defects, the phonon dispersion relation becomes flattened, meaning that within certain frequency ranges, the phonon energy changes very little with respect to the wave vector.^{R5} The associated DOS (**Figure R8e**) displays a quite similar feature with H-type GPG except for a strong intensity in the 1400–1750 cm^{-1} region (highlighted in green), suggesting a different vibrational signature aroused by defects and special structure of p-phenyl in the Z-type GPG. The magnified dispersion in this frequency window (**Figure R8f**) reveals the optical branches and a G^+ mode near 1610 cm^{-1} , with slightly altered dispersion compared to the H-type.

These results highlight the phonon-mode similarities and differences between the two structural configurations. While both share common features associated with sp^2 -hybridized carbon vibrations, subtle distinctions in the dispersion relations and DOS reflect the underlying structural symmetries and interlayer interactions unique to each GPG type. Such vibrational fingerprints provide valuable insights into the lattice dynamics and potential phonon-mediated phenomena in these systems.

Figure R8: (a) The complete phonon dispersion; (b) the DOS; and (c) the corresponding phonon dispersion with frequency from 1400 to 1750 cm^{-1} of H-type GPG; (d) The complete phonon dispersion; (e) the DOS; and (f) The corresponding phonon dispersion with frequency from 1400 to 1750 cm^{-1} of Z-type GPG

Reference:

R4 Novoselov, Kostya S., et al. "Electric field effect in atomically thin carbon films." *Science* 306.5696 (2004): 666-669. <https://www.science.org/doi/10.1126/science.1102896>

R5 Yan, Xingxu, et al. "Single-defect phonons imaged by electron microscopy." *Nature* 589.7840 (2021): 65-69. <https://www.nature.com/articles/s41586-020-03049-y>

To elucidate the vibrational distinctions introduced by the GPG (graphene–phenyl–graphene) architecture, we compared the phonon density of states (DOS) for H-type and Z-type GPG structures with that of pristine graphene, as illustrated in **Figure R9**. The phonon spectra of both H- and Z-type GPG structures retain the fundamental characteristics of graphene, exhibiting no imaginary frequencies across the entire Brillouin zone—that is, all vibrational modes have positive frequencies ($\omega > 0$). This absence of imaginary modes indicates that the structures are dynamically stable within the harmonic approximation. Such stability implies strong mechanical robustness, with the lattice capable of resisting small perturbations without collapsing or undergoing spontaneous structural transitions. Thermodynamically, the phonon contributions to the free energy (e.g., Helmholtz free energy) remain real and well-defined at finite temperatures, enabling reliable evaluation of thermodynamic properties such as heat capacity and entropy. Moreover, dynamic stability typically correlates with experimental feasibility, suggesting that these structures can be synthesized and remain stable under ambient conditions without decomposition or reconstruction.

The phonon DOS of pristine graphene (right panel) exhibits well-known features: a sharp low-frequency acoustic peak, a mid-frequency valley, and a distinct high-frequency peak near 1600 cm^{-1} associated

with the in-plane E_{2g} mode of sp^2 -bonded carbon atoms. These signatures reflect the high symmetry and two-dimensional nature of the graphene lattice. In contrast, both H-type (left panel) and Z-type (middle panel) GPG structures exhibit significantly broadened and redistributed phonon states across the frequency spectrum. This modification arises from the incorporation of p-phenyl linkers, which introduce additional vibrational modes and disrupt the pristine symmetry of graphene. For the H-type GPG, the DOS shows enhanced intensity in the mid-frequency range (600–1300 cm^{-1}) and a broadened feature near 1600 cm^{-1} . The appearance of multiple pronounced peaks indicates increased phonon scattering channels and potential vibrational coupling between graphene layers and phenyl rings. The Z-type GPG displays a similar but more diffuse distribution of states, particularly in the mid-to-high frequency range. Compared to the H-type, the Z-type DOS is less sharply defined near 1600 cm^{-1} , suggesting greater vibrational delocalization and structural complexity due to the rotational asymmetry of the phenyl linkers.

Overall, both GPG configurations demonstrate a clear departure from the discrete and symmetric phonon characteristics of graphene, underscoring the role of interlayer chemical bonding in altering phonon transport and thermal properties. These vibrational fingerprints not only reflect structural distinctions but also have implications for phonon-mediated processes such as thermal conductivity, phonon scattering, and energy dissipation in layered hybrid materials.

Figure R9: The compare of phonon dispersion for H-type GPG, Z-type GPG, and graphene^{R6}

References

- R6** Gale, J. D. & Rohl, A. L. The General Utility Lattice Program (GULP). *Molecular Simulation* **29**, 291-341 (2003). <https://doi.org/10.1080/0892702031000104887>

The phonon dispersion curves have been added to **Figure 1d** and discussed in the revised manuscript.

The following text has been added to the main manuscript:

Figure 1. (d) The phonon density of states and dispersion profiles of Z- and H-type GPG allotropes.

“To elucidate the vibrational characteristics of the two GPG allotropes, we calculated the phonon density of states (DOS) and the phonon dispersion relations for both H-type and Z-type configurations, as presented in Figure 1d. The DOS plots for the H-type and Z-type GPG structures reveal rich vibrational features spanning the entire frequency spectrum. In the H-type structure, a prominent broadening of phonon states is observed in the frequency window between approximately 1400 and 1750 cm^{-1} (region X), where localized vibrational modes associated with the phenyl linkers are likely to emerge. This spectral congestion suggests enhanced phonon–phonon scattering and mode hybridization due to reduced symmetry and strong interfacial coupling between the phenyl units and the graphene layers. In contrast, the Z-type GPG also exhibits a distribution of vibrational states in the same spectral range (region Y), albeit with subtle differences in the peak positions and intensity distribution. These variations reflect the altered stacking geometry and phenyl rotation in the Z-type structure, which modulate the vibrational couplings and interlayer interactions. Phonon dispersion relations further support these observations. The H-type configuration displays noticeable band folding and mode clustering within the 1400–1750 cm^{-1} range (panel x), indicating strong zone-folding effects and symmetry-lowering phenomena. Conversely, the Z-type dispersion (panel y) features a smoother band structure in the same frequency range, but with distinct mode splitting, highlighting the influence of torsional asymmetry and interlayer bonding heterogeneity. The vibrational landscape of GPG is highly sensitive to geometric configuration. The phononic signatures of the H- and Z-type GPG allotropes not only distinguish their structural motifs but also provide insight into their potential thermal transport and vibrational coupling behaviours.”

Comment–5: *Regarding the large discrepancy between the order of change of average carbon energy (AEPC) and free energy (FE) in Figure S3, a more in-depth discussion is recommended. In particular, the low AEPC but very high FE of the C0 structure in Figure S3 may be due to its special electron localization or bond length stress distribution. This phenomenon should be explained by more theoretical analysis or supplementary calculations.*

Response to Comments–5: Thanks for your comments. The discrepancy between average energy per carbon atom (AEPC) and formation energy (FE) in Figure S3 is indeed noteworthy and warranted deeper analysis. To clarify, the FE presented here refers to *formation energy*, not Helmholtz or Gibbs free energy. It is calculated based on the energy difference between the initial and final states, and a positive FE indicates that the formation of the final structure is thermodynamically unfavourable without external energy input.

While AEPC values across the structures vary only slightly (from -8.3 to -8.5 eV), the formation energy more accurately reflects the propensity of a structure to form spontaneously. The C0 structure exhibits a significantly high FE, indicating that although it may be formed under certain conditions, its formation pathway and thermodynamic trend are energetically disfavoured.

To further understand this behaviour, we conducted additional electronic structure analyses, as shown in **Figure R10**. These reveal that localized electron density and bond strain around defect regions contribute to the energy penalty, explaining the high FE. Moreover, localized π -electrons in the planar regions of both graphene and the p-phenyl units induce electrostatic repulsion, further increasing the energy cost compared to delocalized π systems in other GPG configurations.

Figure R10: The top view of charge distribution for (a) graphene layer of C0 structure, (b) p-phenyl group of C0 structure, and (c) side view charge distribution for C0 structure.

These insights have been incorporated into **Figure S5** of the revised supporting information, with a more detailed discussion now provided as follows.

“Figure S5 provides charge density visualizations elucidating the high formation energy (FE) of the C0 structure. The top and side views reveal significant electron localization and bond strain, particularly around the p-phenyl groups and graphene defect regions, which likely contribute to the energetic penalty during formation. This supports the explanation that localized π -electron repulsion and structural stress are responsible for the discrepancy between AEPC and FE.”

Comment–6: *The manuscript mentions molecular modifications such as vinyl, hydroxyethyl, amide, p-phenyl and biphenyl, but lacks comparative data on their specific effects. It is recommended to supplement comparative data on the charge transfer ability, electronic properties and interlayer adjustment ability of these molecules to clarify the reason for choosing p-phenyl. The effects of these molecular modifications on the performance of GPG should be demonstrated through experimental data or calculation results to prove the rationality of the choice of p-phenyl.*

Response to Comments–6: Thanks for your comments. We have supplemented our manuscript with comparative data on different molecular modifications, including vinyl, hydroxyethyl, amide, and biphenyl linkers. Most of them are neither stable nor able to form fine chemical bonds within the graphene layers. Only the biphenyl group is quite stable which we are working on the biphenyl linker and will report soon.

During structural optimisation, the vinyl and hydroxyethyl linkers fail to converge, whereas the amide linkers reach a converged geometry but retain a linked bond. Therefore, from an experimental perspective, it is unfavourable to synthesize the vinyl, hydroxyethyl, and amide linkers in graphene due to their instability or structural non-integrity. However, we believe that we could be successful in preparing the graphene-biphenyl-graphene (GBG) soon for which a brief model and charge of GBG are presented in the column.

Our results indicate that, although the biphenyl-linker-modified graphene yields a convergent and integrated structure, the electrons on the biphenyl linker remain localized rather than delocalized through π - π conjugation as observed in p-phenyl systems, potentially limiting electronic conductivity. Another concern is, the biphenyl linker introduces a significant interlayer distance, resulting in electrochemical behaviour characteristic of surface ion adsorption—favourable for supercapacitors but

less ideal for battery applications. For these reasons, this structure was not included in the current study but will be explored in future work. The relevant structures and energy profiles are provided in **Table R1**.

Table R1. Comparative data of the structural models and electronic properties for different linkers

Molecular group linker	Chemical formula	CHARG/CHGDIFF	Convergence	Structure
Vinyl	G-CH=CH-G		No	hydroxyethyl	G-CH ₂ -O-CH ₂ -G		No	Amide	G-CO-NH-CH ₂ -G		Yes, 1E-6	biphenyl	G- biphenyl-G		Yes, 1E-6	
We analysed their charge transfer capabilities, electronic effects, and influence on interlayer spacing. Our results demonstrate that p-phenyl provides the best balance between structural integrity and

electronic conductivity. That is the reason we select p-phenyl as a linker to prepare the GPG. The relative comparative data on the charge transfer ability, electronic properties and interlayer adjustment ability of these molecules such as vinyl, hydroxyethyl, amide, and biphenyl are included in the **Supporting Information Table S2**.

Comment–7: The pictures of ball-and-stick models in Figure 2b-2c and Figure 3c are not needed. In my opinion, Figure 1 has provided good model images.

Response to Comments–7: Thanks for your comments. We appreciate the reviewer’s insightful suggestion. In line with the recommendation to streamline the presentation, we have removed the ball-and-stick model images from Figures 2b–2c and 3c, as the structural schematics in Figure 1 provide sufficient context. To enhance the clarity of the revised figures, we have supplemented the high-resolution transmission electron microscopy (HRTEM) images in Figures 2g–i, offering additional experimental support for the structural interpretations, as shown in **Figure R11**.

Figure R11. (a) The schematic for layer-by-layer synthesis of GPG from high quality CVD graphene; (b) The 3D and (c) detailed CHGDIFF of GPG between the layers; (d) HRTEM image of GPG top view; (e) AFM image of GPG top view; (f) The height profile for single/double-layer GPG edge; (g) TEM image of GPG; (h) Enlarged TEM and HRTEM (inserted) images of GPG; (i) HRTEM image and modelling structure of the top view for H-type GPG; (j) Side view of the modelling structure of H-type GPG (the ball-type atoms are visible and highlighted in the top view).

Comment-8: Redundancy of charge distribution difference maps. Figure 2d shows the three-dimensional charge distribution difference of GPG, but the manuscript shows 6 charge distribution maps

repeatedly (Figure 2d-i). Just keep the key images. The charge transfer of different structures should be compared through consistent settings of isosurfaces. In addition, specific charge transfer values should be provided to quantitatively analyze how charge redistribution affects the electronic properties of the material, especially the specific effects of different molecular modifications on charge distribution.

Response to Comments–8: Thank you very much for the insightful feedback. We have carefully revised the manuscript to address the redundancy issue noted in Figures 2d–i. Specifically, we have retained the most representative charge distribution difference maps that highlight the key electronic behaviour differences among the studied structures. The redundant figures have been removed to the supporting information to enhance the clarity and conciseness of the presentation (details are provided in the revised **Figure R11**).

To facilitate a more accurate comparison, we have standardized the isosurface settings across all the presented charge distribution maps, ensuring consistent threshold values and visualization parameters. This standardization allows for a direct and meaningful comparison of charge redistribution across different molecular configurations.

Additionally, to quantitatively assess the impact of molecular modifications on the electronic properties of the material, we have included explicit charge transfer values for the H-type and Z-type GPG structures in the revised manuscript (**Figure R12**). These values were obtained by integrating the charge density differences across the interface regions, offering a clear and quantitative measure of charge redistribution effects. The corresponding data and further analysis are supplemented in the Supporting Information.

Figure R12. Quantitative charge transfer values for the (a) H-type and (b) Z-type GPG structures.

The relative content regarding the quantitative charge transfer values for the H and Z-type GPG is supplemented in the supporting information.

“To further elucidate the impact of molecular configuration on interfacial electronic behaviour, we quantitatively analysed the charge transfer distributions for the H-type and Z-type GPG structures. Figure R12 presents the quantitative charge transfer distributions for the H-type and Z-type GPG structures. In the H-type configuration (Figure R12a), charge transfer is highly localized, with significant electron accumulation (up to $-0.16 e$) at specific atomic sites and pronounced charge transfer at the molecule–electrode interfaces ($+0.13 e$ and $+0.04 e$). This indicates strong electronic coupling and localized interaction between the porphyrin core and the graphene electrodes. By contrast, the Z-type configuration (Figure R12b) exhibits a more symmetric and delocalized charge redistribution, with smaller charge transfer values generally within $\pm 0.12 e$. The reduced charge transfer at the interface ($+0.02 e$) suggests weaker electronic coupling compared to the H-type structure. These findings demonstrate that molecular orientation plays a critical role in modulating the interfacial charge transfer and electronic properties, with the H-type structure favouring localized interactions and the Z-type promoting a more distributed electronic environment.”

We believe these revisions significantly improve the clarity, rigor, and quantitative depth of our work, and we sincerely thank the reviewer for this valuable suggestion.

***Comment–9:** In this paper, the high-resolution transmission electron microscopy (HRTEM) image and transmission electron microscopy (TEM) image of GPG prepared by the method of introducing *p*-phenyl groups both show that the structure is rough and defective. It is also possible to randomly introduce *p*-phenyl groups between graphene layers during the experiments. How many *p*-phenyl groups in GPG samples? These factors can affect the final structure and properties of GPG. Please further elaborate on these issues to clarify how to effectively control the uniformity and stability of the structure during the preparation of GPG.*

Response to Comments–9: Thanks for your comment which is very important. Indeed, as noted, the HRTEM and TEM images indicate that the GPG structure exhibits a certain degree of roughness and defectiveness, which is consistent with the Raman data, and we attribute to the introduction of *p*-phenyl groups. The possibility of random insertion of *p*-phenyl moieties between graphene layers during synthesis is a valid concern that we have taken into consideration.

To address this, we have added a more detailed discussion in the revised manuscript (see **Section 1.3** in **Supporting Information**) regarding the synthetic approach and how it influences the structural characteristics of GPG. Specifically, we employed a controlled functionalization process with a carefully optimized molar ratio of *p*-phenyl precursors, reaction time, and temperature to promote uniform grafting. The *p*-phenyl groups could be controlled by the precursor concentration and *p*-phenyl/graphene ratio roughly (**1:25** by mass). The calculation of molar mass suggests an average incorporation of approximately **1** *p*-phenyl groups per **150** carbon atoms, providing a semi-quantitative estimate of the modification degree and aligns well with modelling structure.

Moreover, we acknowledge that local variations in *p*-phenyl incorporation can lead to structural heterogeneity. To minimize this, we are exploring post-synthesis treatments and alternative functionalization routes that may improve uniformity and structural integrity. We have now included these aspects as part of the discussion on synthesis limitations and optimization strategies in the revised **Supporting Information**.

We appreciate the reviewer's suggestion, which helped us better articulate the synthesis challenges and our efforts toward structural control and reproducibility. We must admit the uniformity of *p*-phenyl groups is hard to be precisely controlled. However, we could qualitatively identify that the H-type dominates the GPG synthesized by CVD graphene while Z-type dominates those from RGO. In most cases, both H and Z types co-exist randomly.

Comment–10:

Some typos in the manuscripts:

- 1) 'dynastically' in the sentence 'The carbon allotrope of GPG is thermally and dynastically stable'
- 2) Figure 2c should be checked in the sentence 'The band structures and DOS of both Z and H type GPG were demonstrated in Figure 2c'.

Response to Comments–10: Thank you for your carefully proofreading. We have revised and typos as follows and double-checked the whole manuscript to avoid any typos.

"The carbon allotrope of GPG is thermally and dynastically stable, verified by Density Functional Theory (DFT) and Molecular Dynamics (MD)..."

*"The band structures and DOS of both Z and H type GPG were demonstrated in **Figure 1c**."*

Response to Reviewer #2

Comment–0:

In this manuscript, the author reports a new allotrope to the nanocarbon family, Graphene-P-phenyl-Graphene (GPG), synthesized by inserting π - π -conjugated p-phenyls into the layers of graphene and connecting the graphene layers by C-C σ bonds. My overall view of the paper is that while the work is of potential interest for publication, there are at least following critical weaknesses that need to be addressed before the work can be considered for publication in Nature Communications. Here are detailed reasons and comments.

Response to Comment–0: We sincerely thank the reviewer for the constructive and detailed feedback. We greatly appreciate your recognition of the potential interest of our work and acknowledge the critical issues raised. In response, we have carefully addressed all the concerns by conducting additional experiments and theoretical analyses as recommended. These efforts have substantially strengthened the manuscript and provided a more comprehensive validation of the GPG structure and its properties. We hope the revised version satisfactorily resolves the concerns raised and demonstrates the robustness and novelty of our findings.

Comment–1:

The author needs to give the information of the area capacity when conducting the battery performance test. It is worth noting that too low an area capacity is of little significance for studying the performance improvement of K-ion battery. It is recommended to explain in this article.

Response to Comment–1: Thank you for the valuable comment. We fully agree that areal capacity is a critical parameter for evaluating battery performance, particularly regarding practical applications. In the revised manuscript, we have added the loading information of the GPG active material, which was controlled at approximately 2 mg cm^{-2} under typical test conditions with an area capacity around 1 mAh cm^{-2} . This areal capacity falls within the common range for laboratory-scale coin cell assembly and performance evaluation. We acknowledge, however, that further improvement in areal capacity is essential for advancing toward practical device applications. As such, we are actively working on increasing the areal capacity in future studies, particularly when transitioning from coin cells to larger-format pouch cells. We have now included this discussion in the revised supporting information to clarify the significance of the areal capacity values used in this study.

“To validate the reproducibility of the electrochemical performances, all the loading of active material GPG on the current collector is around 2 mg cm^{-2} , which falls within the common range for laboratory-scale coin cell assembly and performance evaluation.”

Comment–2:

In order to better illustrate the charging and discharging states state of GPG cycled at the voltage, it is necessary to add the in situ XRD measurements of the sample at 0–3.0 V in this work.

Response to Comment–2: We thank the reviewer for the insightful and constructive suggestion. We fully agree that incorporating *in situ* X-ray diffraction (XRD) measurements under electrochemical operating conditions (0–3.0 V) can provide valuable insights into the structural evolution of the GPG electrode during potassiation and depotassiation. Such real-time characterization is indeed instrumental in visualizing dynamic structural changes and in deepening our understanding of the underlying charge storage mechanisms.

We would like to clarify, however, that the GPG material investigated in this study—particularly the Z-type configuration—exhibits a highly disordered carbon framework with abundant structural defects. As a result, the powder XRD patterns display broad and weak diffraction features, lacking the sharp Bragg reflections typically associated with well-crystalline materials. This inherent structural disorder significantly limits the resolution and interpretability of *in situ* XRD measurements. The low signal-to-noise ratio, compounded by the constraints of the electrochemical cell design and X-ray transparency, poses substantial challenges in extracting definitive structural information during electrochemical cycling.

Nevertheless, motivated by the reviewer's recommendation, we developed a custom *in situ* XRD cell configuration specifically optimized to accommodate the low-crystallinity nature of GPG. These experimental data, now included in the revised manuscript (**Figure S13**), were conducted across a full charge-discharge cycle within the 0–3.0 V potential window. The GPG-based potassium-ion battery was first discharged from the open-circuit potential (OCP, 2.56 V) to 0.01 V to achieve full potassiation, followed by a charging process up to 3.0 V, completing a full cycle of potassiation and depotassiation (**Figure S13a**). Simultaneously, *in situ* XRD scans were recorded in the 2θ range of 5–35°, encompassing key diffraction regions associated with carbon structures and potassium carbide phases (**Figure S13b**).

The *in-situ* data reveal subtle but informative changes during electrochemical cycling. Specifically, we observe small shifts and intensity modulations in the broad carbon-related peak around $2\theta = 22\text{--}24^\circ$, indicating slight rearrangements in the carbon matrix. Notably, during potassiation, a new diffraction peak emerges at $2\theta \approx 17^\circ$, which we tentatively assign to the formation of KC_8 (**Figure S13c**). Additional minor peaks at approximately $2\theta = 25^\circ$ and 27° are likely attributable to other potassium-rich graphite intercalation compounds (KC_x), formed through the interaction of potassium with the disordered carbon framework. Importantly, no discernible intermediate phases such as KC_{36} or KC_{24} —typically observed in graphite—were detected, consistent with the amorphous and defect-rich nature of GPG. The

broadening and left-shifting of the main carbon peak during potassiation suggests interlayer expansion due to potassium insertion, while the increase in peak intensity hints at a partial ordering effect induced by K⁺-GPG interactions.

Upon full depotassiation (charging to 3.0 V), all KC₈ and KC_x-associated peaks disappear, and the main carbon peak shifts back to its original position (**Figure S13d**), indicating a high degree of structural reversibility. These findings confirm the structural reversibility of the GPG material under electrochemical operating conditions, consistent with a capacitive and diffusion-controlled charge storage mechanism.

We are grateful for the reviewer's suggestion, which has led to an important experimental extension of our work. The inclusion of *in situ* XRD provides further evidence supporting the structural dynamics and reversibility of the GPG electrode, reinforcing our proposed charge storage model. We will continue to improve the structural characterization methodologies accordingly.

Figure R13. (a) charge-discharge profile for the potassium ion battery; (b) in situ XRD patterns for the GPG potassium ion battery within potential window of 0-3 V; (c) the in situ XRD peaks related to the formation of KC_8 during charge-discharge processes; (d) the representative XRD patterns for GPG at OCP, full potassiation, and full depotassiation states.

Comment–3:

This work requires the addition of typical Nyquist plots of the battery at various SOC values, which can analyze the stability under the voltage charging conditions.

Response to Comment–3: Thanks for your comments. We sincerely appreciate the reviewer for this valuable suggestion. In response, we have incorporated typical Nyquist plots of the battery at various states of charge (SOC) to further analyse the stability and electrochemical behaviour under different charging conditions. The evolution of the impedance spectra with varying SOC provides critical insights into the charge transfer resistance and overall cell stability during cycling. These results are now included in the revised **Supporting Information** and detailed analysis are shown as the follows in **Figure R14**.

Figure R14. Nyquist plots of the battery at different states of charge (SOC) (0%, 25%, 50%, 75%, and 100%). The observed decrease in the charge transfer resistance (R_{ct}) with increasing SOC demonstrates improved electrochemical kinetics at higher SOC values.

“Figure R14 presents typical Nyquist plots of the battery at various states of charge (SOC), ranging from 0% to 100%. At 0% SOC, the impedance spectrum exhibits a notably large semicircle, indicative of a high charge transfer resistance ($R_{ct} = \sim 600 \Omega$), reflecting sluggish electrochemical kinetics at a deeply discharged state. As the SOC increases, the diameter of the semicircle progressively decreases, highlighting a significant reduction in R_{ct} and suggesting enhanced charge transfer processes. At higher SOC levels (75% and 100%), the impedance response stabilizes ($R_{ct} = \sim 200 \Omega$), demonstrating improved interfacial kinetics and greater electrochemical stability under elevated voltage conditions. These results underscore the favourable evolution of the electrochemical interface as the battery charges, affirming the robustness of the system across different SOC levels.”

Comment–4:

The article mentions that graphene obtained from CVD preparation can also synthesize GPG. It is suggested to provide the electrochemical properties of it. Please explain the reason for the difference in electrochemical properties of GPG obtained from different graphene.

Response to Comment–4: Thanks for your comments. It is a good suggestion. The key difference between GPG synthesized from CVD graphene and from rGO or exfoliated graphene lies in the sample state and structural characteristics. GPG synthesized from CVD graphene typically retains a continuous film form (**Figure R15a**), while rGO- or exfoliated graphene-derived GPG generally presents as powder (**Figure R15b**). Structurally, H-type GPG is dominant in CVD-derived samples, benefitting from fewer defects and highly ordered lattice orientations. In contrast, Z-type GPG is dominant in rGO-derived samples, due to higher defect density and structural disorder.

These structural differences significantly affect their electrochemical behaviours. The CVD graphene-derived GPG mainly exhibits surface adsorption behaviours controlled by capacitive processes (**Figure R15c**), thanks to its high lattice order and consistency. Conversely, GPG powder derived from rGO

shows diffusion-controlled behaviours due to its short-range order and long-range lattice disorder (Figure R15d).

Figure R15: Illustration of (a) CVD GPG film and (b) rGO-derived GPG powder; (c) CVD GPG film after K ion adsorption; (d) rGO-derived GPG powder after K ion intercalation.

Figure R16: CV curves of GPG film and rGO-derived GPG powder as electrode for K ion battery.

We have also supplemented and included the comparative electrochemical CV curves between these two types of GPG (**Figure R16**). The differences in electrochemical performance are attributed to variations in structure and morphology as illustrated in **Figure R15**. The detailed analysis is supplemented in the **Supporting Information** as follows.

*“To further elucidate the impact of the starting graphene material on the electrochemical behavior of GPG, we systematically compared GPG synthesized from CVD-grown graphene films with that prepared from exfoliated graphene powders. As shown in **Figure R16**, the CVD-derived GPG film exhibits notably higher electrical conductivity, which is attributed to its low defect density and highly ordered crystalline structure. In contrast, the exfoliated graphene-derived GPG powder, characterized by a greater number of structural defects and lower long-range order, displays lower conductivity. In terms of electrochemical capacity, the exfoliated GPG powder initially delivers a much higher specific capacitance due to its increased surface area and accessible internal active sites. The CVD GPG film demonstrates a low capacity owing to its continuous film structure that provides only the top layer surface for ion adsorption. This behaviour is primarily attributed to the surface adsorption-dominated, capacitive-controlled charge storage mechanism favored by the highly ordered lattice of the CVD GPG. In contrast, the exfoliated GPG powder exhibits more diffusion-controlled kinetics, with notable oxidation and reduction peaks. These observations highlight the critical role of structural order, defect density, and sample form in governing the electrochemical performance of GPG materials.”*

Response to Reviewer #3

Comment–0:

This work reported a new type of allotrope by inserting π - π -conjugated p-phenyls into the layers of graphene. Owing to the expanded interlayer spacing, the reported allotrope exhibits high hall mobility and excellent rate performance as well as cycling stability when used as K ion battery anode. However, following issues on the reported allotrope remains to be discussed before publication.

Response to Comment–0: We sincerely appreciate the reviewer's thoughtful assessment and recognition of the novelty of our reported carbon allotrope, Graphene-P-phenyl-Graphene (GPG). We agree that further clarification and discussion are essential to fully support the structural and electrochemical claims presented. In response, we have carefully addressed all the specific concerns raised by incorporating additional theoretical calculations, control experiments, and structural characterizations. We believe the revised manuscript provides a more rigorous and complete account of our findings, and we thank the reviewer again for the insightful comments that helped improve the quality and clarity of our work.

Comment–1:

In the section of experimental verification, the author characterized interlayer spacing and chemical composition of prepared allotrope GPG via AFM, TEM, Raman, FTIR in combination with NMR. Can the author give more direct experimental evidence on the specific configuration of GPG, especially the bonding between P-phenyl and graphene. Will characterizations such as aberration-corrected transmission electron microscopy and noncontact atomic force microscopy (which has been used in Science 2021, 372, 852-856 and Nature 2023, 614, 95-101) be helpful for structural verification?

Response to Comment–1: Thanks for your constructive comments. We believe the aberration-corrected transmission electron microscopy and noncontact atomic force will be helpful for structural verification. To bridge the gap between theory and experiment, firstly we have compared the theoretically predicted H- and Z-type structures with aberration-corrected transmission electron microscopy acquired along the same crystallographic directions. These comparisons show a notable degree of agreement, particularly in atomic alignment and lattice symmetry, supporting the relevance of our models.

The relative content has been supplemented, and **the relative references have been cited in the main text** as follows:

Figure R17: (g) TEM image of GPG; (h) Enlarged TEM and ACTEM (inserted) images of GPG; (i) ACTEM image and modelling structure of the top view for H-type GPG; (j) Side view of the modelling structure of H-type GPG (the ball-type atoms are visible and highlighted in the top view).

“Following the TEM image of GPG synthesized from CVD graphene (Figure R17g), the enlarged HRTEM image (Figure R17h) provides a clearer view of the atomic structure of H-type GPG. The region highlighted by the red square is further magnified in the inset, revealing a well-defined hexagonal atomic lattice. This lattice matches precisely with the top-view structural model of H-type GPG (Figure R17i), where the atomic positions exhibit excellent alignment. The corresponding side-view model (Figure R17j) confirms that the regular hexagonal lattice arises from the top graphene layer (highlighted in pink),

while the atoms observed at the centre and periphery of the structure (in lilac) originate from the upper shoulder atoms of the p-phenyl groups.”

The Z-type configuration dominates the GPG synthesized from oxidized graphene oxide, where structural defects are typically present. The ACTEM top-view image of the Z-type GPG (**Figure R18a**) reveals that while the hexagonal ring structure of the graphene plane is partially retained, it becomes more disordered. A comparison between the defect locations (**Figure R18b**) and the Z-type GPG model (**Figure R18c**) shows a good alignment of atomic positions (**Figure R18d**). Additionally, the QSTEM simulation closely matches the HRTEM image (**Figure R18e, f**). The side view of the Z-type GPG (**Figure R18g**) highlights the top-layer atoms characteristic of this configuration.^{R7}

Figure R18: (a) TEM image of GPG synthesized from graphene oxide precursors; (b) Enlarged HRTEM image and (c) corresponding top-view structural model of Z-type GPG; (d) Overlay of the HRTEM image and the top-view model showing atomic alignment in Z-type GPG; (e) Region of the HRTEM image corresponding to (f) QSTEM simulation; (g) Side-view model of Z-type GPG, with top-layer atoms highlighted (represented as ball-type atoms in the top view). Scale bar: 0.5 Å.

“The HRTEM image on the top view of GPG shows a plane defect derived from reconstruction and deformation of carbon atoms due to the bonding of p-phenyl. The QSTEM simulation well-match the HRTEM image, which was obtained from quantitative simulation of the Z-type GPG model.”

Moreover, we agree that the noncontact atomic force microscopy may be helpful for structural verification. However, in our case, the p-phenyl groups are inside the interlayers of GPG so it is difficult to detect them with noncontact atomic force microscopy which normally sensitive on the surface environment. So it is challenging to observe the atomic structure inside the interlayers. Therefore, as an alternative option, we tested the distance between two-layer of GPG and compared with double layer graphene, verifying that the sub-nanometer spacing do align well with the modelling structures. ^{R8}

Figure R19. Verification of the structure for GPG via CVD graphene. AFM of GO, GO-p-GO, rGO, and GPG: (a) GO, (b) GO-P, (c) GO-P-GO and (d) GPG.

“The synthesis of H type GPG was achieved via CVD graphene precursor since it has less defect. The approach was verified by AFM characterizations. In principle, the layer spacing distance of a dual-layer graphene could be evaluated by the step height at the edge of single-layer and dual-layer graphene. In Figure R18, the layer spacing distance of graphene oxide (GO), GO-P-GO, rGO, and GPG were 0.78 ± 0.12 , 1.15 ± 0.18 , 0.38 ± 0.18 , and 0.59 ± 0.10 nm, respectively, which align well with these modelling structures.”

References

R7 Fan, Qitang, et al. "Biphenylene network: A nonbenzenoid carbon allotrope." *Science* 372.6544 (2021): 852-856. <https://www.science.org/doi/abs/10.1126/science.abg4509>

R8 Pan, Fei, et al. "Long-range ordered porous carbons produced from C60." Nature 614.7946 (2023): 95-101. <https://www.nature.com/articles/s41586-022-05532-0>

Comment–2:

Did the author consider dispersion correction in the calculations of formation energy? If did, corresponding descriptions should be supplemented in section 1.5 of supplementary materials.

Response to Comment–2: Yes, exactly, the dispersion correction in the calculations of formation energy was considered. We have supplemented the corresponding descriptions in **Section 1.5** of **Supporting Information** as follows.

“The strongly constrained and appropriately normed (SCAN) meta-GGA with the revised Vydrovvan Voorhis no-local correlation functional (SCAN+rVV10) is utilized for the dispersion correction.^{R9} This approach yields superior predictions of both energetic and structural properties for many types of bonding, addressing limitations that many non-empirical semi-local functionals struggle to encompass.

R10, R11, R12”

References

- R9** Peng, H., Yang, Z.-H., Perdew, J. P. & Sun, J. Versatile van der Waals Density Functional Based on a Meta-Generalized Gradient Approximation. *Physical Review X* 6, 041005 (2016). <https://doi.org/10.1103/PhysRevX.6.041005>
- R10** Deng, X., Dai, S., Zhang, Z. & Luo, B. Geometric optimisation and structural properties of organic–inorganic halide perovskites from van der Waals density functional theory calculations. *Molecular Physics*, e2403670 (2024). <https://doi.org/10.1080/00268976.2024.2403670>
- R11** Emrem, B., Kempt, R., Finzel, K. & Heine, T. London Dispersion-Corrected Density Functionals

*Applied to van der Waals Stacked Layered Materials: Validation of Structure, Energy, and Electronic Properties. Advanced Theory and Simulations 5, 2200055 (2022).
<https://doi.org/https://doi.org/10.1002/adts.202200055>*

R12 *Luo, B. et al. Novel atomic-scale graphene metamaterials with broadband electromagnetic wave absorption and ultra-high elastic modulus. Carbon 196, 146-153 (2022).
<https://doi.org/10.1016/j.carbon.2022.04.065>*

Comment–3:

The author screened the suitable bridge molecules only by comparing the formation energies of different configurations. However, it can be seen that the formation of configuration A2, B2 and C2 in Figure S3 needs to experience chemical reaction processes such as removing NH₂ group. What about the kinetic energy barrier during these processes? In this regard, the energy barrier during GPG formation is also an important indicator which should be considered, especially for configurations with close formation energy (such as configuration B2 and C2).

Response to Comment–3: Thanks for your comments which is very considerable. The formation of configuration A2, B2 and C2 do requires to experience several chemical reaction steps which is shown below.

The initial reactions represent a **diazotization reaction (Step 1)** followed by **further diazotization (Step 2)** of a diamine compound. Here's the steps of the transformation:

Step 1 – Diazotization (Low Temperature, 0–5 °C):

One of the amino groups is converted into a diazonium group. This reaction is exothermic due to the formation of a more stable diazonium ion and the release of gases like nitrogen (if further reacted).

Step 2 – Second Diazotization:

The second amino group undergoes diazotization. This step is also exothermic for the same reasons.

Figure R21. The reaction energy profiles for diazotisation of p-Phenylenediamine

The entire **diazotization reaction** process is **exothermic**. Diazotization of aromatic amines is energetically favourable, and the energy released is mainly due to the formation of stable diazonium salts and byproducts like water and sodium chloride.

Regarding the formation of configuration A2, B2 and C2 in Figure S3, they conducted the following reaction steps.

Step 3 – formation of GPG intermediators:

Step 4 – formation of GPG configuration:

The **Step 3** and **Step 4** for A2, B2 and C2 need to experience chemical reaction processes of removing N_2^+Cl^- groups which may experience kinetic energy barrier. According to your suggestion, we have performed transition state calculations to determine the kinetic barriers associated with GPG formation,

particularly for configurations requiring bond rearrangements. These results, now included in **Figure R20**, provide insights into the feasibility of different molecular modifications.

Figure R20. The reaction energy profiles for the formation of GPG

We conclude that all the processes (**Step 1, 2, 3, and 4**) are energetically favourable and proceed without kinetic barriers. Among configurations with comparable formation energies—such as B2 and C2—both may form concurrently. However, the B2 configuration (Z-type GPG) is thermodynamically more stable, in agreement with experimental observations.

Comment-4:

As to the scalable synthesis of GPG, is the bonding between P-phenyl and graphene same as that in GPG prepared from graphene. It is known that the carbon atom in P-phenyl can also bond with the oxygen atom besides carbon atom in graphene oxide. In this regard, precise structure of GPG produced from scalable synthesis process should be verified.

Response to Comment–4: Thanks for your comments. We thank the reviewer for raising this important point regarding the structural integrity of GPG obtained through scalable synthesis. It is indeed known that P-phenyl moieties can potentially form bonds with oxygen-containing functional groups in graphene oxide, leading to the formation of P-phenyl–O–graphene linkages. However, our previous theoretical calculations (**Table R2**) indicate that such oxygen-mediated bonds are significantly less stable than direct C–C linkages formed in GPG. During the high-temperature thermal treatment under an inert atmosphere, these oxygen-containing groups are largely removed, making the formation of stable interlayer linkages through oxygen unlikely.

Table R2. Comparative data of the structural models and electronic properties for oxygen-mediated linkages.

Molecular group linker	Chemical formula	CHARG/CHGDIFF	Convergence	Structure
hydroxyethyl	G-CH ₂ -O- CH ₂ -G		No	amide	G-CO-NH- CH ₂ -G		Yes, 1E-6	
To validate the precise structure of GPG from the scalable synthesis route, we performed extensive structural characterization. The results consistently indicate that the GPG produced from scalable synthesis process is predominantly composed of Z-type GPG frameworks, accompanied by a certain level of rational defects. The modelling structure is illustrated as follows:

Figure R22: (a) The GPG-V1 model with single defect; (b) CHGDIFF image of GPG-V1 model; (c) GPG-V2 model with dual defects; (d) CHGDIFF image of GPG-V2 model

These findings are supported by both modelling and experimental data, as illustrated in the following structural comparisons, showing strong agreement between the predicted configurations and the observed atomic arrangements.

Figure R23: (a) TEM image of GPG synthesized from graphene oxide precursors; (b) Enlarged HRTEM image and (c) corresponding top-view structural model of Z-type GPG; (d) Overlay of the HRTEM image and the top-view model showing atomic alignment in Z-type GPG; (e) Region of the HRTEM image and the top-view model showing atomic alignment in Z-type GPG; (f) Region of the HRTEM image and the top-view model showing atomic alignment in Z-type GPG; (g) 3D model of Z-type GPG

corresponding to (f) QSTEM simulation; (g) Side-view model of Z-type GPG, with top-layer atoms highlighted (represented as ball-type atoms in the top view). Scale bar: 0.5 Å.

All the relative contents have been supplemented and highlighted in the revised **Manuscript** and **Supporting Information**.

Comment–5:

For the electrochemical performance evaluation (Page 12) as K ion battery, contributions from diffusion and capacitive controlled behavior are suggested to be analyzed (such as Figure 6 in Adv. Energy Mater. 2023, 13, 2300442 and Figure 3 in Adv. Energy Mater. 2021, 11, 2002981).

Response to Comment–5: Thanks for your comments which is very helpful. We have included a quantitative analysis of the diffusion and capacitive-controlled contributions to charge storage, following methodologies from recent literature. The relative content has been revised in the **Supporting Information** and **the relative references have been cited in the main text** as follows:

Figure R24: (a) CV curves of GPG at scan rate of 0.1, 0.2, 0.5, 1, 2, and 5 mV s⁻¹; (b) The plots of log(i) with respect to log(v) at specific peak currents with linear fitting for Oxidation and Reduction processes; (c) CV profile of GPG at the scan rate of 0.1 mV s⁻¹ (shaded region shows the calculated capacitive contribution). (d) Contribution percentages of diffusion and capacitive-controlled processes at different scan rates.

“To distinguish between diffusion-controlled and capacitive contributions to the total current response at a fixed potential, the following relationship was employed: $i = k_1 v^{1/2} + k_2 v$, which can be rearranged as $i/v^{1/2} = k_1 + k_2 v^{1/2}$. Here, k_1 and k_2 are constants, with $k_1 v^{1/2}$ representing the diffusion-controlled component and $k_2 v$ corresponding to the capacitive contribution. ^{R13, R14} Cyclic voltammetry (CV)

measurements were performed at scan rates of 0.1, 0.2, 0.5, 1, 2, and 5 mV s⁻¹ (Figure R24a). The peak current (i_p) followed the power-law relationship $i_p = av^b$, where v is the scan rate and a and b are fitting parameters.^{R15} The extracted b -values (~0.75 for both anodic and cathodic peaks; Figure R24b) indicate a mixed charge storage mechanism involving both diffusion and capacitive-controlled processes. At a scan rate of 0.1 mV s⁻¹, the diffusion-controlled contribution was 50.8%, decreasing progressively with increasing scan rate, as capacitive processes became more dominant (Figure R24c). Quantitative analysis based on curve fitting revealed capacitive contributions of 50.8, 56.9, 61.2, 74.0, 86.3, and 91.2% at scan rates of 0.1, 0.2, 0.5, 1, 2, and 5 mV s⁻¹, respectively (Figure R24d)."

Reference

R13 Lee, Jiyoung, et al. "Fluorine - Rich Covalent Organic Framework to Boost Electrochemical Kinetics and Storages of K⁺ Ions for Potassium - Ion Battery." *Advanced Energy Materials* 13.26 (2023): 2300442. <https://doi.org/10.1002/aenm.202300442>

R14 Sun, Fei, et al. "Carboxyl - dominant oxygen rich carbon for improved sodium ion storage: synergistic enhancement of adsorption and intercalation mechanisms." *Advanced Energy Materials* 11.1 (2021): 2002981. <https://doi.org/10.1002/aenm.202002981>

R15 Li, Huanxin, et al. "Ampere-hour-scale soft-package potassium-ion hybrid capacitors enabling 6-minute fast-charging." *Nature communications* 14.1 (2023): 6407. <https://www.nature.com/articles/s41467-023-42108-6>

Table R3. Linear fitting parameters of the plots of $\log(i)$ with respect to $\log(v)$ at oxidation peak current.

Equation	$y = a + b \cdot x$
Plot	Ox
Weight	No Weighting
Intercept	-0.99714 ± 0.00229

Slope	0.76033 ± 0.00383
Residual Sum of Squares	1.16564E-4
Pearson's r	0.99995
R-Square (COD)	0.9999
Adj. R-Square	0.99987

Table R4. Linear fitting parameters of the plots of $\log(i)$ with respect to $\log(v)$ at reduction peak current.

Equation	$y = a + b \cdot x$
Plot	Re
Weight	No Weighting
Intercept	-1.31414 ± 0.00317
Slope	0.82309 ± 0.00529
Residual Sum of Squares	2.2284E-4
Pearson's r	0.99992
R-Square (COD)	0.99983
Adj. R-Square	0.99979

Comment–6:

According to the cycling profile comparison shown in Table S2, the listed studies are all the references published before 2017. Considering the timeliness of the study, recent published work should be supplemented in the comparison.

Response to Comment–6: Thanks for your comments. The cycling profiles in Table S2 have been updated according to the most recently reported references as below (**Table R5**).

Table R5. Cycling profile comparison between our work and some representative recently reported K-ion battery anodes

Electrode	Initial capacity (mAh g⁻¹)	Highest Rate	Cycle number	Remaining capacity (mAh g⁻¹)	Capacity retention (%)	Charge time (s)	Ref.
GPG	~350	210C	20000	~330	98.9	17.1	Our work
HET (Sb _{1.4} Bi _{0.2} Sn _{0.2} Co _{0.1} Mn _{0.1} Te ₃)	308.8	2000 mA g ⁻¹	500	150	48.5		S1
Bi ₂ Se ₃ @rGO@NC/CNT	259.3	5000 mA g ⁻¹	1000	136	68		S2
Se@P-N-C@Mo ₂ C	66	100 mA g ⁻¹	100	40	51		S3
F ₃ P-NOCM	432	5000 mA g ⁻¹	500	384	89		S4
2H-COFs/S	461	3C (900 mA g ⁻¹)	2400	408	100	1200	S5
FePSe ₃ /hC	359	2000 mA g ⁻¹	2000	90	25		S6
a-KNW/C	64	10C (650 mA g ⁻¹)	1000	59	93.2	360	S7
MoS ₂ /HSCB-x	637	10000 mA g ⁻¹	700	369	57.9		S8
S-KNMCO	99	1C	50	42	41	3600	S9
FCM-2	237.2	8C	2000	187.9	79.2	450	S10

References

- [S1] Wang, Z.; Qiao, S.; Ma, M.; Li, T.; Liu, H. K.; Dou, S. X.; Chong, S. High-Entropy Conversion-Alloying anode material for advanced Potassium-Ion batteries. *ACS Nano* **2025**.
- [S2] Wang, Z.; Qiao, S.; Zhao, Y.; Yuan, L.; Li, T.; Chong, S. Multidimensional Encapsulation Geometry Boosting Bismuth Selenide Anode Material with Fast Kinetics for Superior Potassium-Ion Storage. *Journal of Alloys and Compounds* **2025**, 1024, 180329.
- [S3] Cho, S. W.; Choi, H. H.; Senthamaraiannan, T. G.; Lim, D.; Park, G. D.; Cho, C.; Jeong, S. M.; Saroha, R.; Cho, J. S. Hierarchical Porous One-Dimensional N-Doped c Framework Comprising Ultrafine Mo₂C Catalysts for Stable Na/K–Se Batteries: Experimental and Theoretical Investigations. *Chemical Engineering Journal* **2025**, 512, 162456.
- [S4] Luo, Y.-X.; Sun, Z.-H.; Li, Z.-C.; Ma, Y.-M.; Zheng, W.; Bao, Y.; Han, D.-X.; Niu, L. Superior Potassium Storage in Fluorine and Phosphorous-Induced Porous Carbon Nanosheets. *Journal of Power Sources* **2025**, 641, 236898.
- [S5] Chen, Y.-F.; Fang, Y.; Zhu, N.-N.; Luo, X.; Zhu, G.-Y.; Yang, M.; Chen, R.-H.; Zeng, X.; Xiao, J.-M.; Liu, L.; Ning, G.-H.; Bin, D.-S.; Li, D. Multishelled Hollow Covalent Organic Framework Nanospheres for Stable Potassium Storage. *Angewandte Chemie* **2025**. <https://doi.org/10.1002/ange.202424641>.
- [S6] Wu, X.-H.; Chen, B.-C.; Gu, Z.-Q.; Lu, X.; Zhong, H.-Y.; Huang, P.-W.; Zhang, J.; Tan, X.-Y.; Zhao, Y. Unveiling the Structure–Activity Correlation in Iron Phosphorus Trichalcogenide to Realize Enhanced Potassium Ion Storage. *Chemical Engineering Journal* **2025**, 506, 160266. <https://doi.org/10.1016/j.cej.2025.160266>.
- [S7] Tashlanov, M. Yu.; Marshenya, S. N.; Golubnichiy, A. A.; Aksyonov, D. A.; Antipov, E. V.; Fedotov, S. S. Low-Strain, Long-Life and High-Power K-Ion Anode Material Enabled by a Pyrochlore-Type Framework with Facile 3D Isotropic Diffusion. *Journal of Power Sources* **2024**, 629, 236042.
- [S8] Yang, L.; Wang, X.; Zhu, C.; Wang, H.; Shi, J.; Chen, J.; Tian, W.; Zhu, Y.; Huang, M.; Wu, J.; Wang, H. Optimized Few-Layer MoS₂ Confined in Carbon Bowls via Pore Filling and Chemical Bond Enabling Fast Kinetics for High-Rate Potassium Storage. *Chemical Engineering Journal* **2024**, 502, 157821.

[S9] Singh, S. P.; Patel, A.; Tiwari, A.; Yadav, V.; Mishra, R.; Tiwari, R. K.; Singh, R. K. Enhanced Electrochemical Performance of $K_{0.67}[Ni_{0.3}Mn_{0.6}Co_{0.1}]O_2$ as a Cathode Material for Secondary K-Ion Batteries: Improved K-Ion Insertion and Reduced Charge Transfer Barrier. *Surfaces and Interfaces* **2024**, 55, 105316.

[S10] Ren, Q.; Yan, L.; Yu, X.; Lei, W.; Sun, Z.; Hao, R.; Yang, J.; Shi, Z. Construction of Foam-like Carbon Microspheres with Controllable Pseudo-Graphitic Domains: Synergistic Enhancement of K-Ion Adsorption/Intercalation Storage. *Chemical Engineering Journal* **2024**, 499, 156271.